# Intra Order-Preserving Functions for Calibration of Multi-Class Neural Networks

**Amir Rahimi**[*]
ANU, ACRV
amir.rahimi@anu.edu.au

**Amirreza Shaban**[*]
Georgia Tech
ashaban@uw.edu

**Ching-An Cheng**[*]
Microsoft Research
chinganc@microsoft.com

**Richard Hartley**
Google Research, ANU, ACRV
richard.hartley@anu.edu.au

**Byron Boots**
University of Washington
bboots@cs.washington.edu

## Abstract

Predicting calibrated confidence scores for multi-class deep networks is important for avoiding rare but costly mistakes. A common approach is to learn a post-hoc calibration function that transforms the output of the original network into calibrated confidence scores while maintaining the network's accuracy. However, previous post-hoc calibration techniques work only with simple calibration functions, potentially lacking sufficient representation to calibrate the complex function landscape of deep networks. In this work, we aim to learn general post-hoc calibration functions that can preserve the top-$k$ predictions of any deep network. We call this family of functions *intra order-preserving* functions. We propose a new neural network architecture that represents a class of intra order-preserving functions by combining common neural network components. Additionally, we introduce *order-invariant* and *diagonal* sub-families, which can act as regularization for better generalization when the training data size is small. We show the effectiveness of the proposed method across a wide range of datasets and classifiers. Our method outperforms state-of-the-art post-hoc calibration methods, namely temperature scaling and Dirichlet calibration, in several evaluation metrics for the task.

## 1 Introduction

Deep neural networks have demonstrated impressive accuracy in classification tasks, such as image recognition [8, 28] and medical research [10, 3]. These exciting results have recently motivated engineers to adopt deep networks as default components in building decision systems; for example, a multi-class neural network can be treated as a probabilistic predictor and its softmax output can provide the confidence scores of different actions for the downstream decision making pipeline [6, 2, 21]. While this is an intuitive idea, recent research has found that deep networks, despite being accurate, can be overconfident in their predictions, exhibiting high calibration error [20, 7, 11]. In other words, trusting the network's output naively as confidence scores in system design could cause undesired consequences: a serious issue for applications where mistakes are costly, such as medical diagnosis and autonomous driving.

A promising approach to address the miscalibration is to augment a given network with a parameterized calibration function, such as extra learnable layers. This additional component is tuned post-hoc using a held-out calibration dataset, so that the effective full network becomes calibrated [7, 14, 16, 15, 27, 35]. In contrast to usual deep learning, the calibration dataset here is typically

---

[*]Equal Contribution.

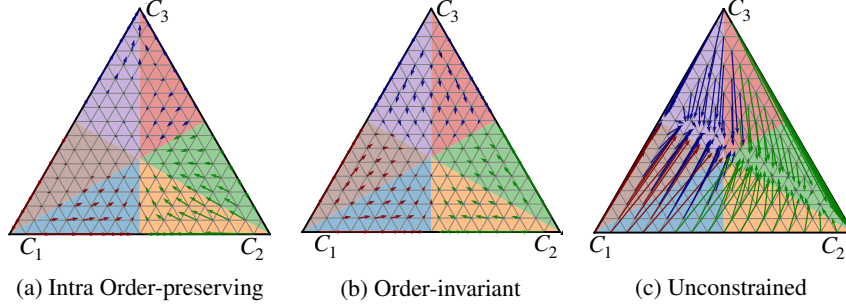

(a) Intra Order-preserving     (b) Order-invariant     (c) Unconstrained

Figure 1: Comparing instances of intra order-preserving and order-invariant family defined on the 2-dimensional unit simplex. Points $C_1 = [1, 0, 0]^\top$, $C_2 = [0, 1, 0]^\top$, $C_3 = [0, 0, 1]^\top$ are the simplex corners. Arrows depict how an input is mapped by each function. Unconstrained function freely maps the input probabilities, intra order-preserving function enforces the outputs to stay within the same colored region as the inputs, and order-invariant function further enforces the vector fields to be the same among all the 6 colored regions as reflected in the symmetry in the visualization.

*small*. Therefore, learning an overly general calibration function can easily overfit and actually reduce the accuracy of the given network [7, 14]. Careful design regularization and parameterization of calibration functions is imperative.

A classical non-parametric technique is isotonic regression [36], which learns a monotonic staircase calibration function with minimal change in the accuracy. But the complexity of non-parametric learning can be too expensive to provide the needed generalization [16, 15]. By contrast, Guo *et al.* [7] proposed to learn a scalar parameter to rescale the original output logits, at the cost of being suboptimal in calibration [20]; see also Section 6. Recently, Kull *et al.* [14] proposed to learn linear transformations of the output logits. While this scheme is more expressive than the temperature scaling above, it does not explore non-linear calibration functions.

In general, a preferable hypothesis space needs to be expressive and, at the same time, provably preserve the accuracy of any given network it calibrates. Limiting the expressivity of calibration functions can be an issue, especially when calibrating deep networks with complicated landscapes.

The main contribution of this paper is introducing a learnable space of functions, called *intra order-preserving* family. Informally speaking, an intra order-preserving function $\mathbf{f} : \mathbb{R}^n \to \mathbb{R}^n$ is a vector-valued function whose output values always share the same ordering as the input values across the $n$ dimensions. For example, if $\mathbf{x} \in \mathbb{R}^n$ is increasing from coordinate 1 to $n$, then so is $\mathbf{f}(\mathbf{x})$. In addition, we introduce order-invariant and diagonal structures, which utilize the shared characteristics between different input dimensions to improve generalization. For illustration, we depict instances of 3-dimensional intra order-preserving and order-invariant functions defined on the unit simplex and compare them to an unconstrained function in Fig. 1. We use arrows to show how inputs on the simplex are mapped by each function. Each colored subset in the simplex denotes a region with the same input order; for example, we have $\mathbf{x}_3 > \mathbf{x}_2 > \mathbf{x}_1$ inside the red region where the subscript $_i$ denotes the $i$th element of a vector. For the intra order-preserving function shown in Fig. 1a arrows stay within the same colored region as the inputs, but the vector fields in two different colored region are independent to each other. Order-invariant function in Fig. 1b further keeps the function permutation invariant, enforcing the vector fields to be the same among all the 6 colored regions (as reflected in the symmetry in Fig. 1b). This property of order-preserving functions significantly reduce the hypothesis space in learning, from the functions on whole simplex to functions on one colored region, for better generalization.

We identify necessary and sufficient conditions for describing intra order-preserving functions, study their differentiability, and propose a novel neural network architecture that can represent complex intra order-preserving function through common neural network components. From practical point of view, we devise a new post-hoc network confidence calibration technique using different intra order-invariant sub-families. Because a post-hoc calibration function keeps the top-$k$ class prediction *if and only if* it is an intra order-preserving function, learning the post-hoc calibration function within the intra order-preserving family presents a solution to the dilemma between accuracy and flexibility faced in the previous approaches. We conduct several experiments to validate the benefits of learning

with these new functions for post-hoc network calibration. The results demonstrate improvement over various calibration performance metrics, compared with the original network, temperature scaling [7], and Dirichlet calibration [14].

## 2  Problem Setup

We address the problem of calibrating neural networks for $n$-class classification. Let define $[n] := \{1, \ldots, n\}$, $\mathcal{Z} \subseteq \mathbb{R}^d$ be the domain, $\mathcal{Y} = [n]$ be the label space, and let $\Delta_n$ denote the $n - 1$ dimensional unit simplex. Suppose we are given a trained probabilistic predictor $\boldsymbol{\phi}_o : \mathbb{R}^d \to \Delta_n$ and a small calibration dataset $\mathcal{D}_c$ of i.i.d. samples drawn from an unknown distribution $\pi$ on $\mathcal{Z} \times \mathcal{Y}$. For simplicity of exposition, we assume that $\boldsymbol{\phi}_o$ can be expressed as the composition $\boldsymbol{\phi}_o =: \mathbf{sm} \circ \mathbf{g}$, with $\mathbf{g} : \mathbb{R}^d \to \mathbb{R}^n$ being a non-probabilistic $n$-way classifier and $\mathbf{sm} : \mathbb{R}^n \to \Delta_n$ being the softmax operator[2], i.e. $\mathbf{sm}_i(\mathbf{x}) = \frac{\exp(\mathbf{x}_i)}{\sum_{j=1}^n \exp(\mathbf{x}_j)}$, for $i \in \mathcal{Y}$, where the subscript $_i$ denotes the $i$th element of a vector. When queried at $\mathbf{z} \in \mathcal{Z}$, the probabilistic predictor $\boldsymbol{\phi}_o$ returns $\arg\max_i \boldsymbol{\phi}_{o,i}(\mathbf{z})$ as the predicted label and $\max_i \boldsymbol{\phi}_{o,i}(\mathbf{z})$ as the associated confidence score. (The top-$k$ prediction is defined similarly.) We say $\mathbf{g}(\mathbf{z})$ is the *logits* of $\mathbf{z}$ with respect to $\boldsymbol{\phi}_o$.

Given $\boldsymbol{\phi}_o$ and $\mathcal{D}_c$, our goal is to learn a post-hoc calibration function $\mathbf{f} : \mathbb{R}^n \to \mathbb{R}^n$ such that the new probabilistic predictor $\boldsymbol{\phi} := \mathbf{sm} \circ \mathbf{f} \circ \mathbf{g}$ is better calibrated *and* keeps the accuracy (or similar performance concepts like top-$k$ accuracy) of the original network $\boldsymbol{\phi}_o$. That is, we want to learn new logits $\mathbf{f}(\mathbf{g}(\mathbf{z}))$ of $\mathbf{z}$. As we will discuss, this task is non-trivial, because while learning $\mathbf{f}$ might improve calibration, doing so could also risk over-fitting to the small dataset $\mathcal{D}_c$ and damaging accuracy. To make this statement more precise, below we first review the definition of perfect calibration [7] and common calibration metrics and then discuss challenges in learning $\mathbf{f}$ with $\mathcal{D}_c$.

**Definition 1.** For a distribution $\pi$ on $\mathcal{Z} \times \mathcal{Y}$ and a probabilistic predictor $\boldsymbol{\psi} : \mathbb{R}^d \to \Delta_n$, let random variables $\mathbf{z} \in \mathcal{Z}$, $y \in \mathcal{Y}$ be distributed according to $\pi$, and define random variables $\hat{y} := \arg\max_i \boldsymbol{\psi}_i(\mathbf{z})$ and $\hat{p} := \boldsymbol{\psi}_{\hat{y}}(\mathbf{z})$. We say $\boldsymbol{\psi}$ is *perfectly calibrated* with respect to $\pi$, if for any $p \in [0, 1]$, it satisfies $\text{Prob}(\hat{y} = y | \hat{p} = p) = p$.

Note that $\mathbf{z}$, $y$, $\hat{y}$ and $\hat{p}$ are correlated random variables. Therefore, Definition 1 essentially means that, if $\boldsymbol{\psi}$ is perfectly calibrated, then for any $p \in [0, 1]$, the true label $y$ and the predicted label $\hat{y}$ match, with a probability exactly $p$ in the events where $\mathbf{z}$ satisfies $\max_i \boldsymbol{\psi}_i(\mathbf{z}) = p$.

In practice, learning a perfectly calibrated predictor is unrealistic, so we need a way to measure the calibration error. A common calibration metric is called Expected Calibration Error (ECE) [23]: $\text{ECE} = \sum_{m=1}^M \frac{|B_m|}{N} |\text{acc}(B_m) - \text{conf}(B_m)|$. This equation is calculated in two steps: First the confidence scores of samples in $\mathcal{D}_c$ are partitioned into $M$ equally spaced bins $\{B_m\}_{m=1}^M$. Second the weighted average of the differences between the average confidence $\text{conf}(B_m) = \frac{1}{|B_m|} \sum_{i \in B_m} \hat{p}^i$ and the accuracy $\text{acc}(B_m) = \frac{1}{|B_m|} \sum_{i \in B_m} \mathbb{1}(y^i = \hat{y}^i)$ in each bin is computed as the ECE metric, where $|B_m|$ denotes the size of bin $B_m$, $\mathbb{1}$ is the indicator function, and the superscript $^i$ indexes the sampled random variable. In addition to ECE, other calibration metrics have also been proposed [7, 25, 1, 17]; e.g., Classwise-ECE [14] and Brier score [1] are proposed as measures of classwise-calibration. All the metrics for measuring calibration have their own pros and cons. Here, we consider the most commonly used metrics for measuring calibration and leave their analysis for future work.

While the calibration metrics above measure the deviation from perfect calibration in Definition 1, they are usually not suitable loss functions for optimizing neural networks, e.g., due to the lack of continuity or non-trivial computation time. Instead, the calibration function $\mathbf{f}$ in $\boldsymbol{\phi} = \mathbf{sm} \circ \mathbf{f} \circ \mathbf{g}$ is often optimized indirectly through a surrogate loss function (e.g. the negative log-likelihood) defined on the held-out calibration dataset $\mathcal{D}_c$ [7].

### 2.1  Importance of Inductive Bias

Unlike regular deep learning scenarios, here the calibration dataset $\mathcal{D}_c$ is relatively small. Therefore, controlling the capacity of the hypothesis space of $\mathbf{f}$ becomes a crucial topic [7, 15, 14]. There is typically a trade-off between preserving accuracy and improving calibration: Learning $\mathbf{f}$ could

improve the calibration performance, but it could also change the decision boundary of $\phi$ from $\phi_o$ decreasing the accuracy. While using simple calibration functions may be applicable when $\phi_o$ has a simple function landscape or is already close to being well calibrated, such a function class might not be sufficient to calibrate modern deep networks with complex decision boundaries as we will show in the experiments in Section 6.

The observation above motivates us to investigate the possibility of learning calibration functions within a hypothesis space that can provably guarantee preserving the accuracy of the original network $\phi_o$. The identification of such functions would address the previous dilemma and give precisely the needed structure to ensure generalization of calibration when the calibration datatset $\mathcal{D}_c$ is small.

## 3 Intra Order-Preserving Functions

In this section, we formally describe this desirable class of functions for post-hoc network calibration. We name them *intra order-preserving functions*. Learning within this family is both necessary and sufficient to keep the top-$k$ accuracy of the original network unchanged. We also study additional function structures on this family (e.g. limiting how different dimensions can interact), which can be used as regularization in learning calibration functions. Last, we discuss a new neural network architecture for representing these functions.

### 3.1 Setup: Sorting and Ranking

We begin by defining sorting functions and ranking in preparation for the formal definition of intra order-preserving functions. Let $\mathbb{P}^n \subset \{0,1\}^{n \times n}$ denote the set of $n \times n$ permutation matrices. Sorting can be viewed as a permutation matrix; Given a vector $\mathbf{x} \in \mathbb{R}^n$, we say $S : \mathbb{R}^n \to \mathbb{P}^n$ is a *sorting function* if $\mathbf{y} = S(\mathbf{x})\mathbf{x}$ satisfies $\mathbf{y}_1 \geq \mathbf{y}_2 \geq \cdots \geq \mathbf{y}_n$. In case there are ties in the input vector $\mathbf{x}$, the sorting matrix can not be uniquely defined. To resolve this, we use a pre-defined *tie breaker* vector which is used as a tie breaking protocol. We say a vector $\mathbf{t} \in \mathbb{R}^n$ is a tie breaker if $\mathbf{t} = P\mathbf{r}$, for some $P \in \mathbb{P}^n$, where $\mathbf{r} = [1, \ldots, n]^\top \in \mathbb{R}^n$. Tie breaker pre-assigns priorities to indices of the input vector and is used to resolve ties. For instance, $\mathbf{S}_1 = \left[\begin{smallmatrix} 1 & 0 \\ 0 & 1 \end{smallmatrix}\right]$ and $\mathbf{S}_2 = \left[\begin{smallmatrix} 0 & 1 \\ 1 & 0 \end{smallmatrix}\right]$ are the unique sorting matrices of input $\mathbf{x} = [0, 0]^\top$ with respect to tie breaker $\mathbf{t}_1 = [1, 2]^\top$ and $\mathbf{t}_2 = [2, 1]^\top$, respectively. We say two vectors $\mathbf{u}, \mathbf{v} \in \mathbb{R}^n$ *share the same ranking* if $S(\mathbf{u}) = S(\mathbf{v})$ for *any* tie breaker $\mathbf{t}$.

### 3.2 Intra Order-Preserving Functions

We define the *intra* order-preserving property with respect to different coordinates of a vector input.

**Definition 2.** We say a function $\mathbf{f} : \mathbb{R}^n \to \mathbb{R}^n$ is *intra order-preserving*, if, for any $\mathbf{x} \in \mathbb{R}^n$, both $\mathbf{x}$ and $\mathbf{f}(\mathbf{x})$ share the same ranking.

The output of an intra order-preserving function $\mathbf{f}(\mathbf{x})$ maintains *all* ties and strict inequalities between elements of the input vector $\mathbf{x}$. Namely, for all $i, j \in [n]$, we have $\mathbf{x}_i > \mathbf{x}_j$ (or $\mathbf{x}_i = \mathbf{x}_j$) if and only if $\mathbf{f}_i(\mathbf{x}) > \mathbf{f}_j(\mathbf{x})$ (or $\mathbf{f}_i(\mathbf{x}) = \mathbf{f}_j(\mathbf{x})$). For example, a simple intra order-preserving function is the temperature scaling $\mathbf{f}(\mathbf{x}) = \mathbf{x}/t$ for some $t > 0$. Another common instance is the softmax operator.

Clearly, applying an intra order-preserving function as the calibration function in $\phi = \mathbf{sm} \circ \mathbf{f} \circ \mathbf{g}$ does not change top-$k$ predictions between $\phi$ and $\phi_o = \mathbf{sm} \circ \mathbf{g}$.

Next, we provide a necessary and sufficient condition for constructing continuous, intra order-invariant functions. This theorem will be later used to design neural network architectures for learning calibration functions. Note that for a vector $\mathbf{v} \in \mathbb{R}^n$ and an upper-triangular matrix of ones $U$, $U\mathbf{v}$ is the *reverse* cumulative sum of $\mathbf{v}$ (i.e. $(U\mathbf{v})_i = \sum_{j=i}^n \mathbf{v}_i$).

**Theorem 1.** *A continuous function* $\mathbf{f} : \mathbb{R}^n \to \mathbb{R}^n$ *is intra order-preserving, if and only if* $\mathbf{f}(\mathbf{x}) = S(\mathbf{x})^{-1} U \mathbf{w}(\mathbf{x})$ *with $U$ being an upper-triangular matrix of ones and* $\mathbf{w} : \mathbb{R}^n \to \mathbb{R}^n$ *being a continuous function such that*

- $\mathbf{w}_i(\mathbf{x}) = 0$, *if* $\mathbf{y}_i = \mathbf{y}_{i+1}$ *and* $i < n$,
- $\mathbf{w}_i(\mathbf{x}) > 0$, *if* $\mathbf{y}_i > \mathbf{y}_{i+1}$ *and* $i < n$,
- $\mathbf{w}_n(\mathbf{x})$ *is arbitrary,*

*where* $\mathbf{y} = S(\mathbf{x})\mathbf{x}$ *is the sorted version of* $\mathbf{x}$.

The proof is deferred to Appendix. Here we provide as sketch as to why Theorem 1 is true. Since $\mathbf{w}_i(\mathbf{x}) \geq 0$ for $i < n$, applying the matrix $U$ on $\mathbf{w}(\mathbf{x})$ results in a sorted vector $U\mathbf{w}(\mathbf{x})$. Thus, applying $S(\mathbf{x})^{-1}$ further on $U\mathbf{w}(\mathbf{x})$ makes sure that $\mathbf{f}(\mathbf{x})$ has the same ordering as the input vector $\mathbf{x}$. The reverse direction can be proved similarly. For the continuity, observe that the sorting function $S(\mathbf{x})$ is piece-wise constant with discontinuities only when there is a tie in the input $\mathbf{x}$. This means that if the corresponding elements in $U\mathbf{w}(\mathbf{x})$ are also equally valued when a tie happens, the discontinuity of the sorting function $S$ does not affect the continuity of $\mathbf{f}$ inherited from $\mathbf{w}$.

### 3.3 Order-invariant and Diagonal Sub-families

Different classes in a classification task typically have shared characteristics. Therefore, calibration functions sharing properties across different classes can work as a suitable inductive bias in learning. Here we use this idea to define two additional structures interesting to intra order-preserving functions: *order-invariant* and *diagonal* properties. Similar to the purpose of the previous section, we will study necessary and sufficient conditions for functions with these properties.

First, we study the concept of order-invariant functions.

**Definition 3.** We say a function $\mathbf{f} : \mathbb{R}^n \to \mathbb{R}^n$ is *order-invariant*, if $\mathbf{f}(P\mathbf{x}) = P\mathbf{f}(\mathbf{x})$ for all $\mathbf{x} \in \mathbb{R}^n$ and permutation matrices $P \in \mathbb{P}^n$.

For an order-invariant function $\mathbf{f}$, when two elements $\mathbf{x}_i$ and $\mathbf{x}_j$ in the input $\mathbf{x}$ are swapped, the corresponding elements $\mathbf{f}_i(\mathbf{x})$ and $\mathbf{f}_j(\mathbf{x})$ in the output $\mathbf{f}(\mathbf{x})$ are also swapped. In this way, the mapping learned for the $i$th class can also be used for the $j$th class. Thus, the order-invariant family shares the calibration function between different classes while allowing the output of each class be a function of all other class predictions.

We characterize in the theorem below the properties of functions that are both intra order-preserving and order-invariant (an instance is the softmax operator). It shows that, to make an intra order-preserving function also order-invariant, we just need to feed the function $\mathbf{w}$ in Theorem 1 with the sorted input $\mathbf{y} = S(\mathbf{x})\mathbf{x}$ instead of $\mathbf{x}$. This scheme makes the learning of $\mathbf{w}$ easier since it always sees sorted vectors (which are a subset of $\mathbb{R}^n$).

**Theorem 2.** *A continuous, intra order-preserving function $\mathbf{f} : \mathbb{R}^n \to \mathbb{R}^n$ is order-invariant, if and only if $\mathbf{f}(\mathbf{x}) = S(\mathbf{x})^{-1}U\mathbf{w}(\mathbf{y})$, where $U$, $\mathbf{w}$, and $\mathbf{y}$ are in Theorem 1.*

Another structure of interest here is the diagonal property.

**Definition 4.** We say a function $\mathbf{f} : \mathbb{R}^n \to \mathbb{R}^n$ is *diagonal*, if $\mathbf{f}(\mathbf{x}) = [f_1(\mathbf{x}_1), \ldots, f_n(\mathbf{x}_n)]$ for $f_i : \mathbb{R} \to \mathbb{R}$ with $i \in [n]$.

In the context of calibration, a diagonal calibration function means that different class predictions do not interact with each other in $\mathbf{f}$. Defining diagonal family is mostly motivated by the success of temperature scaling method [7], which is a linear diagonal intra order-preserving function. Therefore, although diagonal intra order-preserving functions may sound limiting in learning calibration functions, they still represent a useful class of functions.

The next theorem relates diagonal intra order-preserving functions to increasing functions.

**Theorem 3.** *A continuous, intra order-preserving function $\mathbf{f} : \mathbb{R}^n \to \mathbb{R}^n$ is diagonal, if and only if $\mathbf{f}(\mathbf{x}) = [\bar{f}(\mathbf{x}_1), \ldots, \bar{f}(\mathbf{x}_n)]$ for some continuous and increasing function $\bar{f} : \mathbb{R} \to \mathbb{R}$.*

Compared with general diagonal functions, diagonal intra order-preserving automatically implies that the same function $\bar{f}$ is shared across all dimensions. Thus, learning with diagonal intra order-preserving functions benefits from parameter-sharing across different dimensions, which could drastically decrease the number of parameters.

Finally, below we show that functions in this sub-family are also order-invariant and inter order-preserving. Note that inter and intra order-preserving are orthogonal definitions. Inter order-preserving is also an important property for calibration functions, since this property guarantees that $\mathbf{f}_i(\mathbf{x})$ increases with the original class logit $\mathbf{x}_i$. The set diagram in Fig. 2 depicts the relationship among different intra order-preserving families.

**Definition 5.** We say a function $\mathbf{f} : \mathbb{R}^m \to \mathbb{R}^n$ is *inter order-preserving* if, for any $\mathbf{x}, \mathbf{y} \in \mathbb{R}^m$ such that $\mathbf{x} \geq \mathbf{y}$, $\mathbf{f}(\mathbf{x}) \geq \mathbf{f}(\mathbf{y})$, where $\geq$ denotes elementwise comparison.

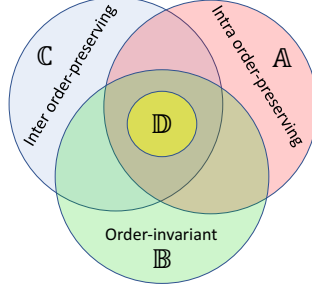

Figure 2: Relationship between different function families. Theorem 1 specifies the intra order-preserving functions $\mathbb{A}$. Theorem 2 specifies the intra order-preserving and order-invariant functions $\mathbb{A} \cap \mathbb{B}$. Theorem 3 specifies the diagonal intra order-preserving functions $\mathbb{D}$. By Corollary 1, these functions are also order-invariant and inter order-preserving i.e. $\mathbb{D} \subseteq \mathbb{A} \cap \mathbb{B} \cap \mathbb{C}$.

**Corollary 1.** *A diagonal, intra order-preserving function is order-invariant and inter order-preserving*

### 3.4 Practical Considerations

Theorems 1 and 2 describe general representations of intra order-preserving functions through a function $\mathbf{w}$ that satisfies certain non-negative constraints. Inspired by these theoretical results, we propose a neural network architecture, Fig. 3, to represent exactly a family of intra order-preserving functions.

The main idea in Fig. 3 is to parameterize $\mathbf{w}$ through a composition of smaller functions. For $i < n$, we set $\mathbf{w}_i(\mathbf{x}) = \sigma(\mathbf{y}_i - \mathbf{y}_{i+1})\mathbf{m}_i(\mathbf{x})$, where $\sigma : \mathbb{R} \to \mathbb{R}$ is a positive function such that $\sigma(a) = 0$ only when $a = 0$, and $\mathbf{m}_i$ is a strictly positive function. It is easy to verify that this parameterization of $\mathbf{w}$ satisfies the requirements on $\mathbf{w}$ in Theorem 1. However, we note that this class of functions cannot represent all possible $\mathbf{w}$ stated in Theorem 1. In general, the speed $\mathbf{w}_i(\mathbf{x})$ converges to 0 can be a function of $\mathbf{x}$, but in the proposed factorization above, the rate of convergence to zero is a function of only two elements $\mathbf{y}_i$ and $\mathbf{y}_{i+1}$. Fortunately, such a limitation does not substantially decrease the expressiveness of $\mathbf{f}$ in practice, because the subspace where $\mathbf{w}_i$ vanishes has zero measure in $\mathbb{R}^n$ (i.e. subspaces where there is at least one tie in $\mathbf{x} \in \mathbb{R}^n$).

By Theorem 1 and Theorem 2, the proposed architecture in Fig. 3 ensures $\mathbf{f}(\mathbf{x})$ is continuous in $\mathbf{x}$ as long as $\sigma(\mathbf{y}_i - \mathbf{y}_{i+1})$ and $\mathbf{m}_i(\mathbf{x})$ are continuous in $\mathbf{x}$. In the appendix, we show that this is true when $\sigma$ and $\mathbf{m}_i$ are continuous functions. Additionally, we prove that when $\sigma$ and $\mathbf{m}$ are continuously differentiable, $\mathbf{f}(\mathbf{x})$ is also directionally differentiable with respect to $\mathbf{x}$. Note that the differentiability to the input is not a requirement to learn the parameters of $\mathbf{m}$ with a first order optimization algorithm which only needs $\mathbf{f}$ to be differentiable with respect to the parameters of $\mathbf{m}$. The latter condition holds in general, since the only potential sources of non-differentiable $\mathbf{f}$, $S(\mathbf{x})^{-1}$ and $\mathbf{y}$ are constant with respect to the parameters of $\mathbf{m}$. Thus, if $\mathbf{m}$ is differentiable with respect to its parameters, $\mathbf{f}$ is also differentiable with respect to the parameters of $\mathbf{m}$.

## 4 Implementation

Given a calibration dataset $\mathcal{D}_c = \{(\mathbf{z}^i, y^i)\}_{i=1}^N$ and a calibration function $\mathbf{f}$ parameterized by some vector $\boldsymbol{\theta}$, we define the empirical calibration loss as $\frac{1}{N} \sum_{i=1}^N \ell(y^i, \mathbf{f}(\mathbf{x}^i)) + \frac{\lambda}{2}||\boldsymbol{\theta}||^2$, where $\mathbf{x}^i = \mathbf{g}(\mathbf{z}^i)$, $\ell : \mathcal{Y} \times \mathbb{R}^n \to \mathbb{R}$ is a classification cost function, and $\lambda \geq 0$ is the regularization weight. Here we follow the calibration literature [30, 7, 14] and use the negative log likelihood (NLL) loss, i.e., $\ell(y, \mathbf{f}(\mathbf{x})) = -\log(\mathbf{sm}_y(\mathbf{f}(\mathbf{x})))$, where $\mathbf{sm}$ is the softmax operator and $\mathbf{sm}_y$ is its $y$th element. We use the NLL loss in all the experiments to study the benefit of learning $\mathbf{f}$ with different structures. The study of other loss functions for calibration [29, 33] is outside the scope of this paper.

To ensure $\mathbf{f}$ is within the intra order-preserving family, we restrict $\mathbf{f}$ to have the structure in Theorem 1 and set $\mathbf{w}_i(\mathbf{x}) = \sigma(\mathbf{y}_i - \mathbf{y}_{i+1})\mathbf{m}(\mathbf{x})$, as described in Section 3.4. We parameterize function $\mathbf{m}$ by a generic multi-layer neural network and utilize the softplus activation $s^+(a) = \log(1 + \exp(a))$

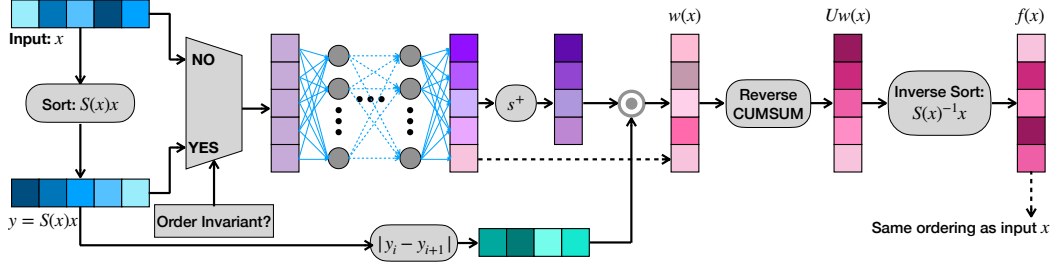

Figure 3: Flow graph of the intra order-preserving function. The vector $\mathbf{x} \in \mathbb{R}^n$ is the input to the graph. Function $\mathbf{m}$ is estimated using a generic multi-layer neural network with non-linear activation for the hidden layers. The input to the network is sorted for learning order-preserving functions. We employ softplus activation function $s^+$ to impose strict positivity constraints.

on the last layer when strict positivity is desired and represent $\sigma$ as $\sigma(a) = |a|$. For example, when $\mathbf{m}_i(\mathbf{x})$ is constant, our architecture recovers the temperature scaling scheme [7].

The order-invariant version in Theorem 2 can be constructed similarly. The only difference is that the neural network that parameterizes $\mathbf{m}$ receives instead the sorted input. Fig. 3 illustrates the architecture of these models.

The diagonal intra order-preserving version in Theorem 3 is formed by learning an increasing function shared across all logit dimensions. We use the official implementation of proposed architecture in [31] that learns monotonic functions with unconstrained neural networks.

## 5 Related Work

Many different post-hoc calibration methods have been studied in the literature [27, 7, 14, 16, 15, 17]. Their main difference is in the parametric family of the calibration function. In Platt scaling [27], scale and shift parameters $a, b \in \mathbb{R}$ are used to transform the scalar logit output $x \in \mathbb{R}$ i.e. $f(x) = ax + b$ of a binary classifier. Temperature scaling [7] is a simple extension of Platt scaling for multi-class calibration in which only a single scalar temperature parameter is learned. Dirichlet calibration [14] allows learning within a richer linear functions family $f(\mathbf{x}) = W\mathbf{x} + \mathbf{b}$, where $W \in \mathbb{R}^{n \times n}$ and $\mathbf{b} \in \mathbb{R}^n$ but the learned calibration function may also change the decision boundary of the original model; Kull *et al.* [14] suggested regularizing the off-diagonal elements of $W$ to avoid overfitting. Similar to our work, the concurrent work in Zhang *et al.* [39] also give special attention to order preserving transformations for calibration. However, their introduced functions are less expressive than the ones presented in this work. Earlier works like isotonic regression [36], histogram binning [35], and Bayesian binning [36] are also post-hoc calibration methods.

In contrast to post-hoc calibration methods, several researches proposed to modify the training process to learn a calibrated network in the first place. Data augmentation methods [30, 34] overcome overfitting by enriching the training data with new artificially generated pseudo data points and labels. Mixup [38] creates pseudo data points by computing the convex combination of randomly sampled pairs. Cutmix [34] uses a more efficient combination algorithm specifically designed for image classification in which two images are combined by overlaying a randomly cropped part of the first image on the second image. In label smoothing [26, 22], the training loss is augmented to penalize high confidence outputs. To discourage overconfident predictions, [29] modifies the original NNL loss by adding a cross-entropy loss term with respect to the uniform distribution. Similarly, [18] adds a calibration regularization to the NLL loss via kernel mean embedding.

Bayesian neural networks [5, 20] derive the uncertainty of the prediction by making stochastic perturbations of the original model. Notably, [5] uses dropout as approximate Bayesian inference. [20] estimates the posterior distribution over the parameters and uses samples from this distribution for Bayesian model averaging. These methods are computationally inefficient since they typically feed each sample to the network multiple times.

## 6 Experiments

We evaluate the performance of intra order-preserving (OP), order-invariant intra order-preserving (OI), and diagonal intra order-preserving (DIAG) families in calibrating the output of various im-

Table 1: *ECE (with $M = 15$ bins) on various image classification datasets and models with different calibration methods. The subscript numbers represent the rank of the corresponding method on the given model/dataset. The accuracy of the uncalibrated model is shown in parentheses. The number in parentheses in DIR, MS, and UNCONSTRAINED methods show the change in accuracy for each method.*

| Dataset | Model | Uncal. | TS | DIR | MS | DIAG | OI | OP | UNCONSTRAINED |
|---|---|---|---|---|---|---|---|---|---|
| CIFAR10 | ResNet 110 | $0.0475_8(93.6\%)$ | $0.0113_5$ | $0.0109_4(-0.1\%)$ | $0.0106_3(-0.1\%)$ | $0.0067_2$ | $\mathbf{0.0061_1}$ | $0.0119_6$ | $0.0170_7(-0.4\%)$ |
| CIFAR10 | Wide ResNet 32 | $0.0451_8(93.9\%)$ | $0.0078_4$ | $0.0084_5(+0.3\%)$ | $0.0073_2(+0.3\%)$ | $0.0136_7$ | $\mathbf{0.0064_1}$ | $0.0077_3$ | $0.0097_6(-0.1\%)$ |
| CIFAR10 | DenseNet 40 | $0.0550_8(92.4\%)$ | $0.0095_2$ | $0.0110_4(+0.1\%)$ | $0.0099_3(+0.1\%)$ | $\mathbf{0.0069_1}$ | $0.0116_5$ | $0.0128_7$ | $0.0125_6(-0.5\%)$ |
| SVHN | ResNet 152 (SD) | $0.0086_8(98.1\%)$ | $0.0061_5$ | $0.0058_3(+0.0\%)$ | $0.0060_4(+0.0\%)$ | $0.0057_2$ | $00.0116_6$ | $0.0118_7$ | $\mathbf{0.0015_1}(+0.0\%)$ |
| CIFAR100 | ResNet 110 | $0.1848_8(71.5\%)$ | $0.0238_2$ | $0.0282_5(+0.2\%)$ | $0.0274_4(+0.1\%)$ | $0.0507_7$ | $\mathbf{0.0119_1}$ | $0.0253_3$ | $0.0346_6(-4.4\%)$ |
| CIFAR100 | Wide ResNet 32 | $0.1878_8(73.8\%)$ | $0.0147_2$ | $0.0189_5(+0.1\%)$ | $0.0258_6(+0.1\%)$ | $0.0172_3$ | $\mathbf{0.0126_1}$ | $0.0173_4$ | $0.0421_7(-6.1\%)$ |
| CIFAR100 | DenseNet 40 | $0.2116_8(70.0\%)$ | $0.0090_2$ | $0.0114_4(+0.1\%)$ | $0.0220_6(+0.4\%)$ | $\mathbf{0.0075_1}$ | $0.0098_3$ | $0.0154_5$ | $0.0990_7(-12.9\%)$ |
| CARS | ResNet 50 (pre) | $0.0239_7(91.3\%)$ | $0.0144_3$ | $0.0243_8(+0.2\%)$ | $0.0186_6(-0.3\%)$ | $0.0105_2$ | $\mathbf{0.0103_1}$ | $0.0185_5$ | $0.0182_4(-3.5\%)$ |
| CARS | ResNet 101 (pre) | $0.0218_7(92.2\%)$ | $0.0165_5$ | $0.0225_8(+0.0\%)$ | $0.0191_6(-0.8\%)$ | $\mathbf{0.0102_1}$ | $0.0135_3$ | $0.0125_2$ | $0.0155_4(-3.9\%)$ |
| CARS | ResNet 101 | $0.0421_8(85.2\%)$ | $0.0301_4$ | $0.0245_3(-0.3\%)$ | $0.0345_6(-1.1\%)$ | $\mathbf{0.0206_1}$ | $0.0323_5$ | $0.0358_7$ | $0.0236_2(-7.0\%)$ |
| BIRDS | ResNet 50 (NTS) | $0.0714_8(87.4\%)$ | $0.0319_5$ | $0.0486_6(-0.2\%)$ | $0.0585_7(-1.1\%)$ | $0.0188_2$ | $\mathbf{0.0172_1}$ | $0.0292_4$ | $0.0276_3(-2.2\%)$ |
| ImageNet | ResNet 152 | $0.0654_7(76.2\%)$ | $0.0208_4$ | $0.0452_5(+0.1\%)$ | $0.0567_6(+0.1\%)$ | $\mathbf{0.0087_1}$ | $0.0109_2$ | $0.0167_3$ | $0.1297_8(-33.4\%)$ |
| ImageNet | DenseNet 161 | $0.0572_7(77.1\%)$ | $0.0198_4$ | $0.0374_5(+0.1\%)$ | $0.0443_6(+0.4\%)$ | $\mathbf{0.0103_1}$ | $0.0123_2$ | $0.0168_3$ | $0.1380_8(-28.1\%)$ |
| ImageNet | PNASNet5 large | $0.0610_7(83.1\%)$ | $0.0713_8$ | $0.0398_6(+0.0\%)$ | $0.0217_4(+0.3\%)$ | $0.0117_2$ | $\mathbf{0.0084_1}$ | $0.0133_3$ | $0.0316_5(-4.8\%)$ |
| Average Relative Error | | $1.00_8$ | $0.42_4$ | $0.49_5$ | $0.50_6$ | $\mathbf{0.27_1}$ | $0.33_2$ | $0.41_3$ | $0.66_7$ |

age classification deep networks and compare their results with the previous post-hoc calibration techniques.

**Datasets.** We use six different datasets: CIFAR-{10,100} [13], SVHN [24], CARS [12], BIRDS [32], and ImageNet [4]. In these datasets, the number of classes vary from 10 to 1000. We evaluate the performance of different post-hoc calibration methods to calibrate ResNet [8], Wide ResNet [37], DenseNet [9], and PNASNet5 [19] networks. We follow the experiment protocol in [14, 16] and use cross validation on the calibration dataset to find the best hyperparameters and architectures for all the methods. Please refer to the Appendix for detailed description of the datasets, pre-trained networks, and hyperparameter tuning.

**Baselines.** We compare the proposed function structures with temperature scaling (TS) [7], Dirichlet calibration with off-diagonal regularization (DIR) [14], and matrix scaling with off-diagonal regularization (MS) [14] as they are the current best performing post-hoc calibration methods. We also present the results of the original uncalibrated models (Uncal.) for comparison. To show the effect of intra order-preserving regularization, we also show the results of applying unconstrained multi-layer neural network without intra order-preserving constraint (UNCONSTRAINED). In cross-validation, we tune the architecture as well as regularization weight of UNCONSTRAINED and order-preserving functions. As we are using the same logits as [14], we report their results directly on CIFAR-10, CIFAR-100, and SVHN. However, since they do not present the results for CARS, BIRDS, and ImageNet datasets, we report the results of their official implementation[3] on these datasets.

**Results.** Table 1 summarizes the results of our calibration methods and other baselines in terms of ECE and presents the average relative error with respect to the uncalibrated model. Overall, DIAG has the lowest average relative error followed by OI among the models and datasets presented in Table 1. OI is the best-performing method in 7 out of 14 experiments including ResNet 110 and Wide ResNet 32 models on CIFAR datasets as well as state-of-the-art PNASNet5 large model. DIAG family's relative average error is half the MS and DIR methods and 15% less compared to Temp. Scaling. Although DIR and MS were able to maintain the accuracy of the original models in most of the cases by imposing off diagonal regularization, order-preserving family could significantly outperform them regarding the ECE metric. Finally, we remark that learning an unconstrained multi-layer neural network does not exhibit a good calibration performance and drastically hurts the accuracy in some datasets as shown in the last column of Table 1.

Fig. 4 illustrates the reliability diagrams of models trained on ResNet 152 (top row) and PNAS-Net5 large (bottom row). Weighted reliability diagrams are also presented to better indicate the differences regarding the ECE metric. Surprisingly, these diagrams show that the uncalibrated PNAS-Net5 large model is underconfident. The differences between the mappings learned by DIAG and temperature scaling on these models are illustrated on the right column. DIAG is capable of learning complex increasing functions while temperature scaling only scales all the logits. Compared with DIR and MS which learn a linear transformation, all intra order-preserving methods can learn non-linear transformations on the logits while decoupling accuracy from calibration of the predictions.

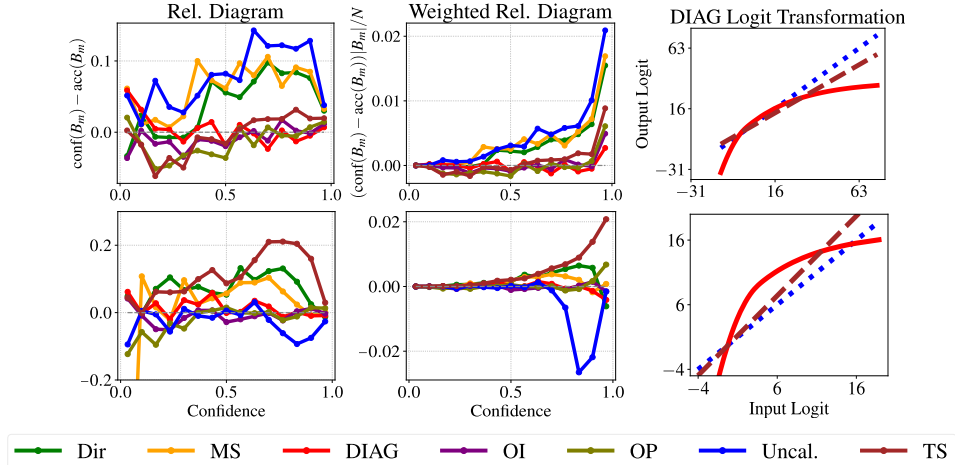

Figure 4: Performance evaluations of ResNet 152 (**Top Row**) and PNASNet5 large (**Bottom Row**) on ImageNet dataset. (**Left**) Reliability diagrams. As suggested by [20] we show the difference between the estimated confidence and accuracy over $M = 15$ bins. The dashed grey lines represent the perfectly calibrated network at $y = 0$. Points above (below) the grey line show overconfident (underconfident) predictions in a bin. (**Middle**) Weighted reliability diagrams where bin values are weighted by data frequency distribution. Since the uncalibrated network has different distances to the perfect calibration in different bins, scaling by a single temperature will lead to a mix of underconfident and overconfident regions. Our order-preserving functions, on the other hand, have more flexibility to reduce the calibration error. (**Right**) Transformation learned by DIAG function compared to temperature scaling and uncalibrated model (identity map).

In addition to ECE, which considers the top prediction, we also measure the NLL, Marginal Calibration Error [17], Classwise-ECE, and Berier score. As it is shown in Table 2, DIAG and OI have the best overall performance in terms of average relative error in most cases, while DIR is the top performing method in Classwise-ECE. Refer to the Appendix for discussions and the performance comparisons over all the datasets.

Table 2: *Average relative error. Each entry shows the relative error compared to the uncalibrated model averaged over all the datasets. The subscripts represent the rank of the corresponding method on the given metric. See the Appendix for per dataset performance comparisons.*

| Evaluation Metric | Uncal. | TS | DIR | MS | DIAG | OI | OP |
|---|---|---|---|---|---|---|---|
| ECE | $1.000_7$ | $0.420_4$ | $0.490_5$ | $0.500_6$ | $\mathbf{0.270}_1$ | $0.330_2$ | $0.410_3$ |
| Debiased ECE [17] | $1.000_7$ | $0.357_3$ | $0.430_6$ | $0.409_5$ | $\mathbf{0.213}_1$ | $0.337_2$ | $0.406_4$ |
| NLL | $1.000_7$ | $0.766_4$ | $0.772_6$ | $0.768_5$ | $\mathbf{0.749}_1$ | $0.751_2$ | $0.765_3$ |
| Marginal Caliration Error [17] | $1.000_7$ | $0.750_3$ | $0.735_2$ | $0.996_6$ | $\mathbf{0.725}_1$ | $0.778_4$ | $0.898_5$ |
| Classwise-ECE | $1.000_7$ | $0.752_6$ | $\mathbf{0.704}_1$ | $0.734_3$ | $0.729_2$ | $0.740_4$ | $0.743_5$ |
| Brier | $1.000_7$ | $0.936_5$ | $0.930_3$ | $0.936_5$ | $\mathbf{0.924}_1$ | $0.929_2$ | $0.931_4$ |

# 7 Conclusion

In this work, we introduce the family of intra order-preserving functions which retain the top-$k$ predictions of any deep network when used as the post-hoc calibration function. We propose a new neural network architecture to represent these functions, and new regularization techniques based on order-invariant and diagonal structures. In short, calibrating neural network with this new family of functions generalizes many existing calibration techniques, with additional flexibility to express the post-hoc calibration function. The experimental results show the importance of learning within the intra order-preserving family as well as support the effectiveness of the proposed regularization in calibrating multiple classifiers on various datasets.

We believe the applications of intra order-preserving family are not limited to network calibration. Other promising domains include, e.g., data compression, depth perception system calibration, and tone-mapping in images where tone-maps need to be monotonic. Exploring the applicability of intra order-preserving functions in other applications is an interesting future direction.

## Broader Impact

Predicting calibrated confidence scores for multi-class deep networks is important for avoiding rare but costly mistakes. Trusting the network's output naively as confidence scores in system design could cause undesired consequences: a serious issue for applications where mistakes are costly, such as medical diagnosis, autonomous driving, suspicious events detection, or stock-market. As an example, in medical diagnosis, it is vital to estimate the chance of a patient being recovered by a certain operation given her/his condition. If the estimation is overconfident/underconfident this will put the life of the patient at risk. Confidence calibrated models would enable integration into downstream decision-making systems, allow machine learning interpretability, and help gain the user trust. While this work focuses primarily on some of the theoretical aspects of the neural network calibration, it also proposes novel techniques to potentially improve broader set of applications where preserving the rank of set of inputs is desired e.g. tone-mapping in images where tone-maps need to be monotonic, depth perception system calibration, and data compression.

We need to remark that our research shows that methods perform differently under various calibration metrics. Unfortunately, discrepancy between different calibration metrics is not well understood and fully explored in the literature. We believe more insights into these inconsistencies would be valuable to the field. We report the performance under different calibration metrics to highlight these differences for the future research. This also means that integrating the proposed work or any other calibration method into decision making systems requires application specific considerations. Other than that, since this work is mostly on the theoretical aspect of improving calibration, we do not foresee any direct negative impacts.

## Acknowledgments and Disclosure of Funding

We would like to thank Antoine Wehenkel for providing helpful instructions for his unconstrained monotonic networks. This research is supported in part by the Australia Research Council Centre of Excellence for Robotics Vision (CE140100016).

## Footnotes

[2]The softmax requirement is not an assumption but for making the notation consistent with the literature. The proposed algorithm can also be applied to the output of general probabilistic predictors.

[3]https://github.com/dirichletcal/experiments_dnn/

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
