[Supplementary Material]

# A  Missing Proofs

## A.1  Proof of Theorem 1, Intra Order-preserving Functions

**Theorem 1.** *A continuous function $\mathbf{f} : \mathbb{R}^n \to \mathbb{R}^n$ is intra order-preserving, if and only if $\mathbf{f}(\mathbf{x}) = S(\mathbf{x})^{-1} U \mathbf{w}(\mathbf{x})$ with $U$ being an upper-triangular matrix of ones and $\mathbf{w} : \mathbb{R}^n \to \mathbb{R}^n$ being a continuous function such that*

- $\mathbf{w}_i(\mathbf{x}) = 0$, *if* $\mathbf{y}_i = \mathbf{y}_{i+1}$ *and* $i < n$,
- $\mathbf{w}_i(\mathbf{x}) > 0$, *if* $\mathbf{y}_i > \mathbf{y}_{i+1}$ *and* $i < n$,
- $\mathbf{w}_n(\mathbf{x})$ *is arbitrary,*

*where $\mathbf{y} = S(\mathbf{x})\mathbf{x}$ is the sorted version of $\mathbf{x}$.*

*Proof of Theorem 1.* ($\to$) For a continuous intra order-preserving function $\mathbf{f}(\mathbf{x})$, let $\mathbf{w}(\mathbf{x}) = U^{-1}S(\mathbf{x})\mathbf{f}(\mathbf{x})$. First we show $\mathbf{w}$ is continuous. Because $\mathbf{f}$ is intra order-preserving, it holds that $S(\mathbf{x}) = S(\mathbf{f}(\mathbf{x}))$. Let $\hat{\mathbf{f}}(\mathbf{x}) := S(\mathbf{f}(\mathbf{x}))\mathbf{f}(\mathbf{x})$ be the sorted version of $\mathbf{f}(\mathbf{x})$. The above implies $\mathbf{w}(\mathbf{x}) = U^{-1}\hat{\mathbf{f}}(\mathbf{x})$. By Lemma 1, we know $\hat{\mathbf{f}}$ is continuous and therefore $\mathbf{w}$ is also continuous.

**Lemma 1.** *Let $\mathbf{f} : \mathbb{R}^n \to \mathbb{R}^n$ be a continuous intra order-preserving function. $S(\mathbf{f}(\mathbf{x}))\mathbf{f}(\mathbf{x})$ is a continuous function.*

Next, we show that $\mathbf{w}$ satisfies the properties listed in Theorem 1. As $\mathbf{w}(\mathbf{x}) = U^{-1}\hat{\mathbf{f}}(\mathbf{x})$, we can equivalently write $\mathbf{w}$ as

$$\mathbf{w}_i(\mathbf{x}) = \begin{cases} \hat{\mathbf{f}}_i(\mathbf{x}) - \hat{\mathbf{f}}_{i+1}(\mathbf{x}) & 1 \le i < n \\ \hat{\mathbf{f}}_n(\mathbf{x}) & i = n. \end{cases}$$

Since $\hat{\mathbf{f}}$ is the sorted version of $\mathbf{f}$, it holds that $\mathbf{w}_i(\mathbf{x}) \ge 0$ for $1 \le i < n$. Also, by the definition of the order-preserving function, $\mathbf{w}_i(\mathbf{x})$ can be zero if and only if $\mathbf{y}_i = \mathbf{y}_{i+1}$, where $\mathbf{y} = S(\mathbf{x})\mathbf{x}$. These two arguments prove the necessary condition.

($\leftarrow$) For a given $\mathbf{w}(\mathbf{x})$ satisfying the condition in the theorem statement, let $\mathbf{v}(\mathbf{x}) = U\mathbf{w}(\mathbf{x})$. Equivalently, we can write $\mathbf{v}_i(\mathbf{x}) = \sum_{j=0}^{n-i} \mathbf{w}_{n-j}(\mathbf{x})$ and $\mathbf{v}_i(\mathbf{x}) - \mathbf{v}_{i+1}(\mathbf{x}) = \mathbf{w}_i(\mathbf{x}), \forall i \in [n]$. By construction of $\mathbf{w}$, one can conclude that $\mathbf{v}(\mathbf{x})$ is a sorted vector where two consecutive elements $\mathbf{v}_i(\mathbf{x})$ and $\mathbf{v}_{i+1}(\mathbf{x})$ are equal if and only if $\mathbf{y}_i = \mathbf{y}_{i+1}$. Therefore, $\mathbf{f}(\mathbf{x}) = S(\mathbf{x})^{-1}\mathbf{v}(\mathbf{x})$ has the same ranking as $\mathbf{x}$. In other words, $\mathbf{f}$ is an intra order-preserving function. The continuity of $\mathbf{f}$ follows from the lemma below and the fact that $\mathbf{v}$ is continuous when $\mathbf{w}$ is continuous. Lemma 2.

**Lemma 2.** *Let $\mathbf{v} : \mathbb{R}^n \to \mathbb{R}^n$ be a continuous function in which $\mathbf{v}_i(\mathbf{x})$ and $\mathbf{v}_{i+1}(\mathbf{x})$ are equal if and only if $\mathbf{y}_i = \mathbf{y}_{i+1}$, where $\mathbf{y} = S(\mathbf{x})\mathbf{x}$. Then $\mathbf{f}(\mathbf{x}) = S(\mathbf{x})^{-1}\mathbf{v}(\mathbf{x})$ is a continuous function.*

∎

### A.1.1  Deferred Proofs of Lemmas

*Proof of Lemma 1.* Let $\mathbb{P}^n = \{P_1, \ldots, P_K\}$ be the finite set of all possible $n \times n$ dimensional permutation matrices. For each $k \in [K]$, define the closed set $\mathbb{N}_k = \{\mathbf{x} : S(\mathbf{x})\mathbf{x} = P_k\mathbf{x}\}$. These sets are convex polyhedrons since each can be defined by a finite set of linear inequalities; in addition, they together form a covering set of $\mathbb{R}^n$. Note that $S(\mathbf{x}) = P_k$ is constant in the interior $int(\mathbb{N}_k)$, but $S(\mathbf{x})$ may change on the boundary $\partial(\mathbb{N}_k)$ which corresponds to points where a tie exists in elements of $\mathbf{x}$ (for such a point $S(\mathbf{x}) \ne P_k$). Nonetheless, by definition of the set $\mathbb{N}_k$, we have $S(\mathbf{x})\mathbf{x} = P_k\mathbf{x}$ for *all* $\mathbf{x} \in \mathbb{N}_k$, which implies that $S(\mathbf{x})$ and $P_k$ can only have different elements for indices where elements of $\mathbf{x}$ are equal.

To prove that $\hat{\mathbf{f}}(\mathbf{x}) := S(\mathbf{f}(\mathbf{x}))\mathbf{f}(\mathbf{x})$ is continuous, we leverage the fact that $\hat{\mathbf{f}}(\mathbf{x}) = S(\mathbf{x})\mathbf{f}(\mathbf{x})$ for intra order-preserving $\mathbf{f}$. We will first show that $\hat{\mathbf{f}}(\mathbf{x}) = P_k\mathbf{f}(\mathbf{x})$ for $\mathbf{x} \in \mathbb{N}_k$ and any $k \in [K]$, which implies $\hat{\mathbf{f}}$ is continuous on $\mathbb{N}_k$ when $\mathbf{f}$ is continuous. To see this, consider an arbitrary $k \in [K]$. For

$\mathbf{x} \in int(\mathbb{N}_k)$ in the interior, we have $S(\mathbf{x}) = P_k$ and therefore $\hat{\mathbf{f}}(\mathbf{x}) = P_k \mathbf{f}(\mathbf{x})$. For $\mathbf{x} \in \partial \mathbb{N}_k$ on the boundary, we have

$$\hat{\mathbf{f}}(\mathbf{x}) = S(\mathbf{x})\mathbf{f}(\mathbf{x}) = P_k\mathbf{f}(\mathbf{x}).$$

The last equality holds because the difference between $S(\mathbf{x})$ and $P_k$ are only in the indices for which elements of $\mathbf{x}$ are equal, and the order-preserving $\mathbf{f}$ preserves exactly the same equalities. Thus, the differences between permutations $S(\mathbf{x})$ and $P_k$ do not reflect in the products $S(\mathbf{x})\mathbf{f}(\mathbf{x})$ and $P_k\mathbf{f}(\mathbf{x})$.

Next, we show that $\hat{\mathbf{f}}(\mathbf{x}) = P_k\mathbf{f}(\mathbf{x}) = P_{k'}\mathbf{f}(\mathbf{x})$ for $\mathbf{x} \in \partial \mathbb{N}_k \cap \partial \mathbb{N}_{k'}$. While $P_k \neq P_{k'}$, the intersection $\partial \mathbb{N}_k \cap \partial \mathbb{N}_{k'}$ contains exactly points $\mathbf{x}$ such that the index differences in $P_k$ and $P_{k'}$ correspond to same value in $\mathbf{x}$. Because $\mathbf{f}$ is order-preserving, by an argument similar to the previous step, we have $P_k\mathbf{f}(\mathbf{x}) = P_{k'}\mathbf{f}(\mathbf{x})$ for $\mathbf{x} \in \partial \mathbb{N}_k \cap \partial \mathbb{N}_{k'}$.

Together these two steps and the fact that $\{\mathbb{N}_k\}$ is covering set on $\mathbb{R}^n$ show that $\hat{\mathbf{f}}$ is a piece-wise continuous function on $\mathbb{R}^n$ when $\mathbf{f}$ is continuous on $\mathbb{R}^n$. ∎

*Proof of Lemma 2.* In order to show the continuity of $\mathbf{f}(\mathbf{x})$, we use a similar argument as in Lemma 1 (see therein for notation definitions). For any $k \in [K]$, it is also trivial to show that $\mathbf{f}$ is continuous over the open set $int(\mathbb{N}_k)$ since $\mathbf{f}(\mathbf{x}) = P_k^{-1}\mathbf{v}(\mathbf{x})$. We use the same argument as Lemma 1 to show it is also a continuous for any point $\mathbf{x} \in \partial(\mathbb{N}_k)$

$$\mathbf{f}(\mathbf{x}) = S(\mathbf{x})^{-1}\mathbf{v}(\mathbf{x}) = P_k^{-1}\mathbf{v}(\mathbf{x}).$$

The last equality holds because $P_k^{-1}$ and $S(\mathbf{x})^{-1}$ can only have different elements among elements of $\mathbf{y} = S(\mathbf{x})\mathbf{x}$ with equal values, and $\mathbf{v}$ preserves exactly these equalities in $\mathbf{y}$. Finally, the proof can be completed by piecing the results of different $\mathbb{N}_k$ together.

∎

## A.2 Proof of Theorem 2, Order-invariant Functions

**Theorem 2.** *A continuous, intra order-preserving function $\mathbf{f} : \mathbb{R}^n \to \mathbb{R}^n$ is order-invariant, if and only if $\mathbf{f}(\mathbf{x}) = S(\mathbf{x})^{-1}U\mathbf{w}(\mathbf{y})$, where $U$, $\mathbf{w}$, and $\mathbf{y}$ are in Theorem 1.*

To prove Theorem 2, we first study the properties of order invariant functions in Appendix A.2.1. We will provide necessary and sufficient conditions to describe order invariant functions, like what we did in Theorem 1 for intra order-preserving functions. Finally, we combine these insights and Theorem 1 to prove Theorem 2 in Appendix A.2.2.

### A.2.1 Properties of Order Invariant Functions

The goal of this section is to prove the below theorem, which characterizes the representation of order invariant functions using the concept of equality-preserving.

**Definition 6.** We say a function $\mathbf{f} : \mathbb{R}^n \to \mathbb{R}^n$ is *equality-preserving*, if $\mathbf{f}_i(\mathbf{x}) = \mathbf{f}_j(\mathbf{x})$ for all $\mathbf{x} \in \mathbb{R}^n$ such that $\mathbf{x}_i = \mathbf{x}_j$ for some $i, j \in [n]$

**Theorem 4.** *A function $\mathbf{f} : \mathbb{R}^n \to \mathbb{R}^n$ is order-invariant, if and only if $\mathbf{f}(\mathbf{x}) = S(\mathbf{x})^{-1}\bar{\mathbf{f}}(S(\mathbf{x})\mathbf{x})$ for some function $\bar{\mathbf{f}} : \mathbb{R}^n \to \mathbb{R}^n$ that is equality-preserving on the domain $\{\mathbf{y} : \mathbf{y} = S(\mathbf{x})\mathbf{x}, \text{ for } \mathbf{x} \in \mathbb{R}^n\}$.*

Theorem 4 shows an order invariant function can be expressed in terms of some equality-preserving function. In fact, every order invariant function is equality-preserving.

**Proposition 1.** *Any order-invariant function $\mathbf{f} : \mathbb{R}^n \to \mathbb{R}^n$ is equality-preserving.*

*Proof.* Let $P_{ij} \in \mathbb{P}^n$ denote the permutation matrix that only swaps $i^{th}$ and $j^{th}$ elements of the input vector; i.e. $\mathbf{y} = P_{ij}\mathbf{x} \Rightarrow \mathbf{y}_i = \mathbf{x}_j, \mathbf{y}_j = \mathbf{x}_i, \mathbf{y}_k = \mathbf{x}_k, \forall \mathbf{x} \in \mathbb{R}^n, i, j, k \in [n]$, and $k \neq i, j$. Thus, for an order-invariant function $\mathbf{f} : \mathbb{R}^n \to \mathbb{R}^n$ and any $\mathbf{x} \in \mathbb{R}^n$ such that $\mathbf{x}_i = \mathbf{x}_j$, we have

$$\mathbf{f}(P_{ij}\mathbf{x}) = P_{ij}\mathbf{f}(\mathbf{x}) \Rightarrow \mathbf{f}_i(P_{ij}\mathbf{x}) = \mathbf{f}_j(\mathbf{x}) \Rightarrow \mathbf{f}_i(\mathbf{x}) = \mathbf{f}_j(\mathbf{x}) \quad (\because P_{ij}\mathbf{x} = \mathbf{x} \text{ for } \mathbf{x} \text{ such that } \mathbf{x}_i = \mathbf{x}_j).$$

∎

We are almost ready to prove Theorem 4. We just need one more technical lemma, whose proof is deferred to the end of this section.

**Lemma 3.** *For any $P \in \mathbb{P}^n$ and an equality-preserving $\mathbf{f} : \mathbb{R}^n \to \mathbb{R}^n$, $S(\mathbf{x})\mathbf{f}(\mathbf{x}) = S(P\mathbf{x})P\mathbf{f}(\mathbf{x})$.*

*Proof of Theorem 4.* ($\to$) For an order-invariant function $\mathbf{f} : \mathbb{R}^n \to \mathbb{R}^n$, we have $\mathbf{f}(P\mathbf{x}) = P\mathbf{f}(\mathbf{x})$ by Definition 3 for any $P \in \mathbb{P}^n$. Take $P = S(\mathbf{x})$. We then have the equality $\mathbf{f}(\mathbf{x}) = S(\mathbf{x})^{-1}\mathbf{f}(S(\mathbf{x})\mathbf{x})$. This is an admissible representation because, by Proposition 1, $\mathbf{f}$ is equality-preserving.

($\leftarrow$) Let $\mathbf{f}(\mathbf{x}) = S(\mathbf{x})^{-1}\bar{\mathbf{f}}(S(\mathbf{x})\mathbf{x})$ for some equality-preserving function $\bar{\mathbf{f}}$. First, because $\bar{\mathbf{f}}$ is equality preserving and $\mathbf{f}$ is constructed through the sorting function $S$, we notice that $\mathbf{f}(\mathbf{x})$ is equality-preserving. Next, we show $\mathbf{f}$ is also order invariant:

$$
\begin{aligned}
\mathbf{f}(P\mathbf{x}) &= S(P\mathbf{x})^{-1}\bar{\mathbf{f}}(S(P\mathbf{x})P\mathbf{x}) \\
&= S(P\mathbf{x})^{-1}\bar{\mathbf{f}}(S(\mathbf{x})\mathbf{x}) && (\because S(P\mathbf{x})P\mathbf{x} = S(\mathbf{x})\mathbf{x} \text{ by choosing } \mathbf{f}(\mathbf{x}) = \mathbf{x} \text{ in Lemma 3}) \\
&= S(P\mathbf{x})^{-1}S(\mathbf{x})\mathbf{f}(\mathbf{x}) && (\because \text{definition of } \mathbf{f}(\mathbf{x})) \\
&= S(P\mathbf{x})^{-1}S(P\mathbf{x})P\mathbf{f}(\mathbf{x}) && (\because \text{Lemma 3}) \\
&= P\mathbf{f}(\mathbf{x}).
\end{aligned}
$$

∎

### A.2.2 Main Proof

*Proof of Theorem 2.* ($\to$) From Theorem 1 we can write $\mathbf{f}(\mathbf{x}) = S(\mathbf{x})^{-1}U\mathbf{w}(\mathbf{x})$. On the other hand, from Theorem 4 we can write $\mathbf{f}(\mathbf{x}) = S(\mathbf{x})^{-1}\bar{\mathbf{f}}(\mathbf{y})$ for some equality-preserving function $\bar{\mathbf{f}}$. Using both we can identify $\mathbf{w}(\mathbf{x}) = U^{-1}\bar{\mathbf{f}}(\mathbf{y})$ which implies that $\mathbf{w}$ is only a function of the sorted input $\mathbf{y}$ and can be equivalently written as $\mathbf{w}(\mathbf{y})$.

($\leftarrow$) For $\mathbf{w}$ with the properties in the theorem statement, the function $\mathbf{f}(x) = S(x)^{-1}U\mathbf{w}(\mathbf{y})$ satisfies the conditions of Theorem 1; therefore $\mathbf{f}$ is intra order-preserving. To show $\mathbf{f}$ is also order-invariant, we write $\mathbf{f}(\mathbf{x}) = S(x)^{-1}\bar{\mathbf{f}}(\mathbf{y})$ where $\bar{\mathbf{f}}(\mathbf{y}) = U\mathbf{w}(\mathbf{y})$. Because $\bar{\mathbf{f}}_i(\mathbf{y}) = \sum_{j=0}^{n-i}\mathbf{w}_{n-j}(\mathbf{x})$, we can derive with the definition of $\mathbf{w}$ that

$$
\mathbf{y}_i = \mathbf{y}_{i+1} \Rightarrow \mathbf{w}_i(\mathbf{y}) = 0 \Rightarrow \bar{\mathbf{f}}_i(\mathbf{x}) = \bar{\mathbf{f}}_{i+1}(\mathbf{x}).
$$

That is, $\bar{\mathbf{f}}(\mathbf{y})$ is equality-preserving on the domain of sorted inputs. Thus, $\mathbf{f}$ is also order-invariant. ∎

### A.2.3 Deferred Proof of Lemmas

*Proof of Lemma 3.* To prove the statement, we first notice a fact that $S(\mathbf{x}) = S(P\mathbf{x})P$, for any $P \in \mathbb{P}^n$ and $\mathbf{x} \in \mathbb{X} := \{\mathbf{x} \in \mathbb{R}^n : \mathbf{x}_i \neq \mathbf{x}_j, \forall i,j \in [n], i \neq j\}$. Therefore, for $\mathbf{x} \in \mathbb{X}$, we have $S(\mathbf{x})\mathbf{f}(\mathbf{x}) = S(P\mathbf{x})P\mathbf{f}(\mathbf{x})$.

Otherwise, consider some $\mathbf{x} \in \mathbb{R}^n \setminus \mathbb{X}$. Without loss of generality[4], we may consider $n > 2$ and $\mathbf{x}$ such that $\mathbf{x}_1 = \mathbf{x}_2 > \mathbf{x}_k$ for all $k > 2$; because $\mathbf{f}$ is equality-preserving, we have $\mathbf{f}_1(\mathbf{x}) = \mathbf{f}_2(\mathbf{x})$.

To prove the desired equality, we will introduce some extra notations. We use subscript $_{i:j}$ to extract contiguous parts of a vector, e.g. $\mathbf{x}_{2:n} = [\mathbf{x}_2, \ldots, \mathbf{x}_n]$ and $\mathbf{f}_{2:n}(\mathbf{x}) = [\mathbf{f}_2(\mathbf{x}), \ldots, \mathbf{f}_n\mathbf{x})]$ (by our construction of $\mathbf{x}$, $\mathbf{x}_{2:n}$ is a vector where each element is unique.) In addition, without loss of generality, suppose $P \in \mathbb{P}^n$ shifts index 1 to some index $i \in [n]$; we define $\bar{P} \in \{0,1\}^{n-1 \times n-1}$ by removing the 1st column and the $i$th row of $P$ (which is also a permutation matrix). Using this notion, we can partition $S(\bar{P}\mathbf{x}_{2:n}) \in \{0,1\}^{n-1 \times n-1}$ as

$$
S(\bar{P}\mathbf{x}_{2:n}) = \begin{bmatrix} B_1 & B_2 \\ B_3 & B_4 \end{bmatrix}
$$

where $B_1 \in \mathbb{R}^{1 \times i-1}$, $B_2 \in \mathbb{R}^{1 \times n-i}$, $B_3 \in \mathbb{R}^{n-2 \times i-1}$, and $B_4 \in \mathbb{R}^{n-2 \times n-i}$. This would imply that $S(P\mathbf{x}) \in \{0,1\}^{n \times n}$ can be written as one of followings

$$
\begin{bmatrix} & e_i^\top & \\ B_1 & 0 & B_2 \\ B_3 & 0 & B_4 \end{bmatrix} \quad \text{or} \quad \begin{bmatrix} B_1 & 0 & B_2 \\ & e_i^\top & \\ B_3 & 0 & B_4 \end{bmatrix} \tag{1}
$$

where $e_i$ is the $i$th canonical basis.

To prove the statement, let $\mathbf{y} = P\mathbf{f}(\mathbf{x})$. By the definition of $\bar{P}$, we can also write $\mathbf{y}$ as

$$\mathbf{y} = \begin{bmatrix} \mathbf{y}_{1:i-1} \\ \mathbf{y}_i \\ \mathbf{y}_{i+1:n} \end{bmatrix} = \begin{bmatrix} (\bar{P}\mathbf{f}_{2:n}(\mathbf{x}))_{1:i-1} \\ \mathbf{f}_1(\mathbf{x}) \\ (\bar{P}\mathbf{f}_{2:n}(\mathbf{x}))_{i:n-1} \end{bmatrix} \tag{2}$$

Let us consider the first case in (1). We have

$$S(P\mathbf{x})P\mathbf{f}(\mathbf{x}) = \begin{bmatrix} \mathbf{y}_i \\ B_1 y_{1:i-1} + B_2 y_{i+1:n} \\ B_3 y_{1:i-1} + B_4 y_{i+1:n} \end{bmatrix} = \begin{bmatrix} \mathbf{y}_i \\ S(\bar{P}\mathbf{x}_{2:n})\bar{P}\mathbf{f}_{2:n}(\mathbf{x}) \end{bmatrix} = \begin{bmatrix} \mathbf{f}_1(\mathbf{x}) \\ S(\mathbf{x}_{2:n})\mathbf{f}_{2:n}(\mathbf{x}) \end{bmatrix} = S(\mathbf{x})\mathbf{f}(\mathbf{x})$$

where the second equality follows from (2), the third from the fact we proved at the beginning for the set $\mathbb{X}$, and the last equality is due to the assumption $\mathbf{x}_1 = \mathbf{x}_2 > \mathbf{x}_k$ and the equality-preserving property that $\mathbf{f}_1(\mathbf{x}) = \mathbf{f}_2(\mathbf{x})$. For the second case in (1), based on the same reasoning above, we can show

$$S(P\mathbf{x})P\mathbf{f}(\mathbf{x}) = \begin{bmatrix} (S(\mathbf{x}_{2:n})\mathbf{f}_{2:n}(\mathbf{x}))_1 \\ \mathbf{f}_1(\mathbf{x}) \\ (S(\mathbf{x}_{2:n})\mathbf{f}_{2:n}(\mathbf{x}))_{2:n-1} \end{bmatrix},$$

Because $\mathbf{x}_1 = \mathbf{x}_2$, we have $(S(\mathbf{x}_{2:n})\mathbf{f}_{2:n}(\mathbf{x}))_1 = \mathbf{f}_1(\mathbf{x}) = \mathbf{f}_2(\mathbf{x})$. Thus, $S(P\mathbf{x})P\mathbf{f}(\mathbf{x}) = S(\mathbf{x})\mathbf{x}$. ∎

### A.3 Proof of Theorem 3, Diagonal Functions

**Theorem 3.** *A continuous, intra order-preserving function $\mathbf{f} : \mathbb{R}^n \to \mathbb{R}^n$ is diagonal, if and only if $\mathbf{f}(\mathbf{x}) = [\bar{f}(\mathbf{x}_1), \ldots, \bar{f}(\mathbf{x}_n)]$ for some continuous and increasing function $\bar{f} : \mathbb{R} \to \mathbb{R}$.*

We first prove some properties of *diagonal* intra order-preserving functions, which will be used to prove Theorem 3.

**Proposition 2.** *Any intra order-preserving function $\mathbf{f} : \mathbb{R}^n \to \mathbb{R}^n$ is equality-preserving.*

*Proof.* This can be seen directly from the definition of intra order-preserving functions. ∎

**Corollary 2.** *The following statements are equivalent*

1. *A function $\mathbf{f} : \mathbb{R}^n \to \mathbb{R}^n$ is diagonal and equality-preserving.*

2. *$\mathbf{f}(\mathbf{x}) = [\bar{f}(\mathbf{x}_1), \ldots, \bar{f}(\mathbf{x}_n)]$ for some $\bar{f} : \mathbb{R} \to \mathbb{R}$.*

3. *A function $\mathbf{f} : \mathbb{R}^n \to \mathbb{R}^n$ is diagonal and order-invariant.*

*Proof.* $(1 \to 2)$ Let $\mathbf{f}(\mathbf{x}) = [f_1(\mathbf{x}_1), \ldots, f_n(\mathbf{x}_n)]$ be a diagonal and equality-preserving function. One can conclude that $\mathbf{f}_1(x) = \cdots = \mathbf{f}_n(x)$ for all $x \in \mathbb{R}$.

$(2 \to 3)$ Let $\mathbf{u} = P\mathbf{x}$ for some permutation matrix $P \in \mathbb{P}^n$. Then $\mathbf{f}(P\mathbf{x}) = [\bar{f}(\mathbf{u}_1), \ldots, \bar{f}(\mathbf{u}_n)] = P[\bar{f}(\mathbf{x}_1), \ldots, \bar{f}(\mathbf{x}_n)] = P\mathbf{f}(\mathbf{x})$.

$(3 \to 1)$ True by Proposition 1. ∎

*Proof of Theorem 3.* $(\to)$ By Proposition 2, an intra order-preserving function $\mathbf{f}$ is also equality-preserving. Therefore, by Corollary 2 it can be represented in the form $\mathbf{f}(\mathbf{x}) = [\bar{f}(\mathbf{x}_1), \ldots, \bar{f}(\mathbf{x}_n)]$ for some $\bar{f} : \mathbb{R} \to \mathbb{R}$. Furthermore, because $\mathbf{f}(\mathbf{x})$ is intra order-preserving, for any $\mathbf{x} \in \mathbb{R}^n$ with $\mathbf{x}_1 > \mathbf{x}_2$, it satisfies $\mathbf{f}_1(\mathbf{x}) > \mathbf{f}_2(\mathbf{x})$; that is, $\bar{f}(\mathbf{x}_1) > \bar{f}(\mathbf{x}_2)$. Therefore, $\bar{f}$ is an increasing function. Continuity is inherited naturally.

$(\leftarrow)$ Because $\mathbf{f}_i(\mathbf{x}) = \bar{f}(\mathbf{x}_i)$ and $\bar{f}$ is an increasing function, it follows that $\mathbf{f}$ is intra order-preserving

$$\mathbf{x}_i = \mathbf{x}_j \Rightarrow \mathbf{f}_i(\mathbf{x}) = \mathbf{f}_j(\mathbf{x}) \qquad \text{and} \qquad \mathbf{x}_i > \mathbf{x}_j \Rightarrow \mathbf{f}_i(\mathbf{x}) > \mathbf{f}_j(\mathbf{x}).$$

∎

Finally, we prove that diagonal intra order-preserving functions are also order-invariant. This fact was mentioned in the paper without a proof.

**Corollary 3.** *A diagonal intra order-preserving function is also order-invariant.*

*Proof.* Intra order-preserving functions are equality-preserving by Proposition 2. By Corollary 2 an diagonal equality-preserving function is order-invariant. ∎

# B Continuity and Differentiability of the Proposed Architecture

In this section, we discuss properties of the function $\mathbf{f}(\mathbf{x}) = S(\mathbf{x})^{-1} U D(\mathbf{y}) \mathbf{m}(\mathbf{x})$. In order to learn the parameters of $\mathbf{m}$ with a first order optimization algorithm, it is important for $\mathbf{f}$ to be differentiable with respect to the parameters of $\mathbf{m}$. This condition holds in general, since the only potential sources of non-differentiable $\mathbf{f}$, $S(\mathbf{x})^{-1}$ and $\mathbf{y}$ are constant with respect to the parameters of $\mathbf{m}$. Thus, if $\mathbf{m}$ is differentiable with respect to its parameters, $\mathbf{f}$ is also differentiable with respect to the parameters of $\mathbf{m}$.

Next, we discuss continuity and differentiability of $\mathbf{f}(\mathbf{x})$ with respect the *input* $\mathbf{x}$. These properties are important when the input to function $f$ is first processed by a trainable function $\mathbf{g}$ (i.e. the final output is computed as $\mathbf{f} \circ \mathbf{g}(\mathbf{x})$). This is not the case in post-hoc calibration considered in the paper, since the classifier $\mathbf{g}$ here is not being trained in the calibration phase.

We show below that when $\mathbf{w}(\mathbf{x}) = D(\mathbf{y}) \mathbf{m}(\mathbf{x})$ satisfies the requirements in Theorem 1, the function $\mathbf{f}(\mathbf{x}) = S(\mathbf{x})^{-1} U D(\mathbf{y}) \mathbf{m}(\mathbf{x})$ is a continuous intra order-preserving function.

**Corollary 4.** *Let $\sigma : \mathbb{R} \to \mathbb{R}$ be a continuous function where $\sigma(0) = 0$ and strictly positive on $\mathbb{R} \setminus \{0\}$, and let $\mathbf{m}$ be a continuous function where $\mathbf{m}_i(\mathbf{x}) > 0$ for $i < n$, and arbitrary for $\mathbf{m}_d(\mathbf{y})$. Let $D(\mathbf{y})$ denote a diagonal matrix with entries $D_{ii} = \sigma(\mathbf{y}_i - \mathbf{y}_{i+1})$ for $i < n$ and $D_{nn} = 1$. Then $\mathbf{w}(\mathbf{x}) = D(\mathbf{y}) \mathbf{m}(\mathbf{x})$ is a continuous function and satisfies the following conditions*

- $\mathbf{w}_i(\mathbf{x}) = 0$, *for $i < n$ and $\mathbf{y}_i = \mathbf{y}_{i+1}$*

- $\mathbf{w}_i(\mathbf{x}) > 0$, *for $i < n$ and $\mathbf{y}_i > \mathbf{y}_{i+1}$*

- $\mathbf{w}_n(\mathbf{x})$ *is arbitrary,*

*where $\mathbf{y} = S(\mathbf{x}) \mathbf{x}$ is the sorted version of $\mathbf{x}$.*

*Proof.* First, because $\mathbf{y} = S(\mathbf{x}) \mathbf{x}$ is a continuous function (by Lemma 1 with $\mathbf{f}(\mathbf{x}) = \mathbf{x}$), $\mathbf{w}(\mathbf{x}) = D(\mathbf{y}) \mathbf{m}(\mathbf{x})$ is also a continuous function. Second, because $\|\mathbf{x}\| < \infty$, we have $\mathbf{m}(\mathbf{x}) < \infty$ due to continuity. Therefore, it follows that $\mathbf{w}_i(\mathbf{x}) = \sigma(\mathbf{y}_i - \mathbf{y}_{i+1}) \mathbf{m}_i(\mathbf{x})$ satisfies all the listed conditions. ∎

To understand the differentiability of $\mathbf{f}$, we first see that $\mathbf{f}$ may not be differentiable at a point where there is a tie among some elements of the input vector.

**Corollary 5.** *For $\mathbf{w}$ in Corollary 4, there exists differentiable functions $\mathbf{m}$ and $\sigma$ such that $\mathbf{f}(\mathbf{x}) = S(\mathbf{x})^{-1} U \mathbf{w}(\mathbf{x})$ is not differentiable globally on $\mathbb{R}^n$.*

*Proof.* For the counter example, let $\mathbf{m} : \mathbb{R}^3 \to \mathbb{R}^3$ be a constant function $\mathbf{m}(\mathbf{x}) = [1, 1, 1]^\top$, and $\sigma(a) = a^2$. It is easy to verify that they both satisfy the conditions in Corollary 4 and are differentiable. We show that the partial derivative $\frac{\partial \mathbf{f}_1(\mathbf{x})}{\partial \mathbf{x}_3}$ does not exists at $\mathbf{x} = [2, 1, 1]^\top$. With few simple steps one could see $\mathbf{f}_1(\mathbf{x} + \alpha \mathbf{e}_3)$ for $\alpha \in (-\infty, 1]$ is

$$\mathbf{f}_1(\mathbf{x} + \alpha \mathbf{e}_3) = \begin{cases} \sigma(1) + \sigma(-\alpha) + 1 & \alpha \leq 0 \\ \sigma(1 - \alpha) + \sigma(\alpha) + 1 & 0 < \alpha \leq 1 \end{cases} \tag{3}$$

Though this function is continuous, the left and right derivatives are not equal at $\alpha = 0$ so the function is not differentiable at $\mathbf{x} = [2, 1, 1]^\top$. ∎

The above example shows that $\mathbf{f}$ may not be differentiable for tied inputs. On the other hand, it is straightforward to see function $\mathbf{f}$ is differentiable at points where there is no tie. More precisely, for the points with tie in the input vector, we show the function $\mathbf{f}$ is B-differentiable, which is a weaker condition than the usual (Frechét) differentiability.

**Definition 7.** [2] A function $\mathbf{f} : \mathbb{R}^n \to \mathbb{R}^m$ is said to be *B(ouligand)-differentiable* at a point $\mathbf{x} \in \mathbb{R}^n$, if $\mathbf{f}$ is Lipschitz continuous in the neighborhood of $\mathbf{x}$ and directionally differentiable at $\mathbf{x}$.

**Proposition 3.** *For* $\mathbf{f} : \mathbb{R}^n \to \mathbb{R}^n$ *in Theorem 1, let* $\mathbf{w}(\mathbf{x})$ *be as defined in Corollary 4. If* $\sigma$ *and* $\mathbf{m}$ *are continuously differentiable, then* $\mathbf{f}$ *is B-differentiable on* $\mathbb{R}^n$.

*Proof.* Let $\mathbb{P}^n = \{P_1, \ldots, P_K\}$ be the finite set of all possible $n \times n$ dimensional permutation matrices. For each $k \in [K]$, define the closed set $\mathbb{N}_k = \{\mathbf{x} : S(\mathbf{x})\mathbf{x} = P_k\mathbf{x}\}$. These sets are convex polyhedrons since each can be defined by a finite set of linear inequalities; in addition, they together form a covering set of $\mathbb{R}^n$.

If there is no tie in elements of vector $\mathbf{x}$, then $\mathbf{x} \in int(\mathbb{N}_k)$ for some $k \in [K]$. Since the sorting function $S(\mathbf{x})$ has the constant value $P_k$ in a small enough neighborhood of $\mathbf{x}$, the function $\mathbf{f}$ is continuously differentiable (and therefore B-differentiable) at $\mathbf{x}$.

Next we show that, for any point $\mathbf{x} \in \mathbb{R}^n$ with some tied elements, the directional derivative of $\mathbf{f}$ along an arbitrary direction $\mathbf{d} \in \mathbb{R}^n$ exists. For such $\mathbf{x}$ and $\mathbf{d}$, there exists a $k \in [K]$ and a small enough $\delta > 0$ such that $\mathbf{x}, \mathbf{x} + \epsilon\mathbf{d} \in \mathbb{N}_k$ for all $0 \le \epsilon \le \delta$. Therefore, we have $\mathbf{f}(\mathbf{x}') = \hat{\mathbf{f}}(\mathbf{x}')$ for all $\mathbf{x}' \in [\mathbf{x}, \mathbf{x} + \delta\mathbf{d}]$, where $\hat{\mathbf{f}}_k(\mathbf{x}) = P_k^{-1}UD(P_k\mathbf{x})\mathbf{m}(\mathbf{x})$. Let $\hat{\mathbf{f}}_k'(\mathbf{x}; \mathbf{d})$ denote the directional derivative of $\hat{\mathbf{f}}_k$ at $\mathbf{x}$ along $\mathbf{d}$. By the equality of $\hat{\mathbf{f}}_k$ and $\mathbf{f}$ in $[\mathbf{x}, \mathbf{x} + \delta\mathbf{d}]$, we conclude that the directional derivative $\mathbf{f}'(\mathbf{x}; \mathbf{d})$ exists and is equal to $\hat{\mathbf{f}}'(\mathbf{x}; \mathbf{d})$.

Finally, we note that $\mathbf{f}$ is Lipschitz continuous, since it is composed by pieces of Lipschitz continuous functions $\hat{\mathbf{f}}_k$ for $k \in [K]$ (implied by the continuous differentiability assumption on $\sigma$ and $\mathbf{m}$). Thus, $\mathbf{f}$ is B-differentiable. ∎

## C  Learning Increasing Functions

We follow the implementation of [10] for learning increasing functions in the diagonal subfamily. The idea is to learn an increasing function $\bar{f}(x) : \mathbb{R} \to \mathbb{R}$ using a neural network, which can be realized by learning a strictly positive function $\bar{f}'(x)$ and a bias $\bar{f}(0) \in \mathbb{R}$ and constructing the desired function $\bar{f}$ by the integral $\bar{f}(x) = \int_0^x \bar{f}'(t)dt + \bar{f}(0)$. In implementation, the derivative $\bar{f}'$ is modeled by a generic neural network and the positiveness is enforced by using a proper activation function in the last layer. In the forward computation, the integral is approximated numerically using Clenshaw-Curtis quadrature [1] and the backward pass is performed by Leibniz integral rule to reduce memory footprint. We the use official implementation of the algorithm provided by [10].

## D  Datasets, Hyperparameters, and Architecture Selection

The size of the calibration and the test datasets, as well as the number of classes for each dataset, are shown in Table 3. We note that the calibration sets sizes are the same as the previous methods [3, 5].

Table 3: Statistics of the Evaluation Datasets.

| Dataset | #classes | Calibration set size | Test dataset size |
|---------|----------|----------------------|-------------------|
| CIFAR-10 | 10 | 5000 | 10000 |
| SVHN | 10 | 6000 | 26032 |
| CIFAR-100 | 100 | 5000 | 10000 |
| CARS | 196 | 4020 | 4020 |
| BIRDS | 200 | 2897 | 2897 |
| ImageNet | 1000 | 25000 | 25000 |

Table 4: Hyperparameters learned by cross validation. For DIAG, OI, OP, and UNCONSTRAINED we show the network architectures learned by cross validation. The number of units in each layer are represented by a sequence of numbers, e.g. $(10, 20, 30, 40)$ represents a network with 10 input units, 20 and 30 units in the first and second hidden layers, respectively, and 40 output units. We perform multi-fold cross-validation and select the architecture with lowest NLL on validation set.

| Dataset | Model | DIAG | OI | OP | UNCONSTRAINED |
|---|---|---|---|---|---|
| CIFAR10 | ResNet 110 | (1,10,10,1) | (10,150,150,10) | (10,2,2,10) | (10,500,10) |
| CIFAR10 | Wide ResNet 32 | (1,2,2,1) | (10,10,10,10) | (10,2,2,10) | (10,150,150,10) |
| CIFAR10 | DenseNet 40 | (1,2,2,1) | (10,50,50,100 | (10,2,2,10) | (10,150,150,10) |
| SVHN | ResNet 152 (SD) | (1,20,20,1) | (10,10,10,10) | (10,50,50,10) | (10,500,10) |
| CIFAR100 | ResNet 110 | (1,10,10,1) | (100,100,100,100) | (100,150,150,100) | (100,500,100) |
| CIFAR100 | Wide ResNet 32 | (1,1,1) | (100,100,100,100) | (100,2,2,100) | (100,500,100) |
| CIFAR100 | DenseNet 40 | (1,1,1) | (100,10,10,100) | (100,2,2,100) | (100,500,500,100) |
| CARS | ResNet 50 (pre) | (1,50,1) | (196,10,196) | (196,2,2,196) | (196,500,196) |
| CARS | ResNet 101 (pre) | (1,20,20,1) | (196,100,100,196) | (196,20,20,196) | (196,500,196) |
| CARS | ResNet 101 | (1,50,1) | (196,50,50,196) | (196,100,100,196) | (196,500,196) |
| BIRDS | ResNet 50 (NTS) | (1,50,50,1) | (200,150,150,200) | (200,50,50,200) | (200,500,200) |
| ImageNet | ResNet 152 | (1,10,10,1) | (1000,150,150,1000) | (1000,2,2,1000) | (1000,150,1000) |
| ImageNet | DenseNet 161 | (1,10,1) | (1000,100,100,1000) | (1000,2,2,1000) | (1000,150,1000) |
| ImageNet | PNASNet5 large | (1,20,20,1) | (1000,50,50,1000) | (1000,100,100,1000) | (1000,100,1000) |

We follow the experiment protocol in [5] and use cross validation on the calibration set to find the best hyperparameters and architectures for all the methods. We found that [5] have improved their performance via averaging output predictions of models trained on different folds. We follow the same approach to have fair comparisons. Our criteria for selecting the best architecture is the NLL value. We perform 3 fold cross validation for ImageNet and 5 folds for all the other datasets. We limit our architecture to fully connected networks and vary the number of hidden layers as well as the size of each layer. We allow networks with up to 3 hidden layers in all the experiments. In CIFAR-10, SVHN, and CIFAR-100 with fewer classes, we test networks with $\{1, 2, 10, 20, 50, 100, 150\}$ units per layer and for the larger CARS, BIRDS, and ImageNet datasets, we allow a wider range of $\{2, 10, 20, 50, 100, 150, 500\}$ units per layer. We use the similar number of units for all the hidden layers to reduce the search space. We use ReLU activation for all middle hidden layers and Softplus on the last layer when strict positivity is desired. We utilize L-BFGS [7] for small scale optimization problems when the computational resources allow (temperature scaling and diagonal intra order-preserving (DIAG) methods on CIFAR and SVHN datasets) and use Adam [4] optimizer for other experiments. Table 4 summarize cross validation learned hyperparameter for each method.

Although the functions learned in Table 4 are more complicated than linear transformations used in the baselines, they are not too complex to slow down computation as the calibration network size is negligible compared to the backbone network used in the experiments. In our experiments, all methods take less than $0.5$ milliseconds/sample in forward path and their differences are negligible.

We use the pre-computed logits of these networks provided by [5] for CIFAR, SVHN, and ImageNet with DenseNet and ResNet [5]. In addition, we use the publicly available state-of-the-art models for PNASNet5-large and ResNet50 NTSNet [11] [6] for ImageNet and BIRDS datasets, respectively. Furthermore, we trained different ResNet type models on CARS dataset using the standard pytorch training script. The ResNet models with (pre) are initialized with pre-trained ImageNet weights. We will release these models for future research.

The effect of weight regularization on different metrics for MS and DIR methods is illustrated in Fig. 5. This shows that simply regularizing the off diagonal elements of a linear layer has limited expressiveness to achieve good calibration especially in the case that number of classes is large.

# E   More Experiments and Discussions

**Reliability Diagrams.** In Fig. 4 of the paper, we show the reliablity diagrams and diagnoal functions leanred by TS and DIAG in ResNet 152 and PNASNet5 large on ImageNet dataset. Fig. 6 and 7 illustrate the reliability diagrams for different calibration algorithms in all the models. In general

Figure 5: Accuracy, ECE, and NLL plots in MS and DIR for ResNet 152 on ImageNet with different regularization weights. In the plots, x-axis shows the log scale regularization and y-axis shows the accuracy, ECE, and NLL of different methods, respectively. The value of the bias regularizer is found by cross validation and kept constant for visualization purpose. Changing the bias regularizer has little effect on the final shape of the plots.

Table 5: *Scores and rankings of different methods for Brier.*

| Dataset | Model | Uncal. | TS | DIR | MS | DIAG | OI | OP |
|---------|-------|--------|-----|-----|-----|------|-----|-----|
| CIFAR10 | ResNet 110 | $0.01102_7$ | $0.00979_6$ | $0.00977_5$ | $0.00976_4$ | $0.00967_2$ | $\textbf{0.00963}_1$ | $0.00975_3$ |
| CIFAR10 | Wide ResNet 32 | $0.01047_7$ | $0.00924_4$ | $\textbf{0.00888}_1$ | $0.00889_2$ | $0.00926_6$ | $0.00921_3$ | $0.00925_5$ |
| CIFAR10 | DenseNet 40 | $0.01274_7$ | $0.01100_4$ | $0.01100_4$ | $\textbf{0.01097}_1$ | $0.01100_4$ | $0.01110_6$ | $0.01099_3$ |
| SVHN | ResNet 152 (SD) | $0.00297_6$ | $\textbf{0.00291}_1$ | $0.00293_3$ | $0.00298_7$ | $0.00292_2$ | $0.00293_3$ | $0.00296_5$ |
| CIFAR100 | ResNet 110 | $0.00453_7$ | $0.00392_4$ | $0.00391_3$ | $0.00391_3$ | $0.00393_5$ | $\textbf{0.00389}_1$ | $0.00390_2$ |
| CIFAR100 | Wide ResNet 32 | $0.00432_7$ | $0.00355_4$ | $0.00354_2$ | $\textbf{0.00351}_1$ | $0.00355_4$ | $0.00354_2$ | $0.00355_4$ |
| CIFAR100 | DenseNet 40 | $0.00491_7$ | $0.00401_3$ | $\textbf{0.00400}_1$ | $\textbf{0.00400}_1$ | $0.00401_3$ | $0.00401_3$ | $0.00402_6$ |
| CARS | ResNet 50 (pretrained) | $0.000667_5$ | $0.000666_4$ | $0.000663_2$ | $0.000679_7$ | $\textbf{0.000661}_1$ | $0.000664_3$ | $0.000674_6$ |
| CARS | ResNet 101 (pretrained) | $0.000626_6$ | $0.000625_5$ | $0.000623_3$ | $0.000655_7$ | $0.000622_2$ | $\textbf{0.000620}_1$ | $0.000623_3$ |
| CARS | ResNet 101 | $0.001131_6$ | $0.001129_5$ | $0.001123_3$ | $0.001154_7$ | $\textbf{0.001118}_1$ | $0.001123_3$ | $0.001119_2$ |
| BIRDS | ResNet 50 (NTSNet) | $0.001035_6$ | $0.000995_5$ | $0.000988_4$ | $0.001040_7$ | $0.000977_3$ | $\textbf{0.000972}_1$ | $0.000974_2$ |
| ImageNet | ResNet 152 | $0.000338_7$ | $0.000332_4$ | $0.000333_6$ | $0.000332_4$ | $\textbf{0.000329}_1$ | $0.000330_2$ | $0.000331_3$ |
| ImageNet | DenseNet 161 | $0.000325_7$ | $0.000321_4$ | $0.000321_4$ | $\textbf{0.000318}_1$ | $0.000319_2$ | $0.000320_3$ | $0.000321_4$ |
| ImageNet | PNASNet5 large | $0.000255_6$ | $0.000261_7$ | $0.000252_5$ | $0.000247_3$ | $0.000245_2$ | $\textbf{0.000244}_1$ | $0.000248_4$ |
| Average Relative Error | | $1.000_7$ | $0.936_5$ | $0.930_3$ | $0.936_5$ | $\textbf{0.924}_1$ | $0.929_2$ | $0.931_4$ |

DIAG method outperforms other methods in calibration in most of the regions. OP and OI methods also achieve good calibration performance on this dataset and are slightly better than temperature scaling, while MS and DIR methods do not reduce the calibration error as much.

**Calibration Set Size.** In this experiment, we gradually increase the calibration set size from $10\%$ to $100\%$ of its original size to create smaller calibration subsets. Then, for each calibration subset, we train different post-hoc calibration methods and measure their accuracy, NLL, and ECE. The results are illustrated in Fig. 8. In overall, the performance of non intra order-preserving methods, i.e. DIR and MS, are more sensitive to the size of the calibration set while intra order-preserving methods maintain the accuracy and are more stable in terms of NLL and ECE.

**Brier Score, NLL, and Classwise-ECE.** As shown in Table 5, our OI is the best method in 5 out 14 models with respect to the Brier score. MS also wins in 4 models. However, it performs poorly on CARS and BIRDS datasets. Our DIAG has the best average relative error. Overall, both OI and DIAG perform well on this metric. The DIR is the third best method on this metric and is slightly worse than OI in average relative error.

Results of different methods regarding the NLL metric are shown in Table 6. MS is the best method when the number of classes is less than or equal to 100 on this metric. Its performance degrades as the number of classes grows. This is typically due to the excessive number of parameters introduced by this method. Surprisingly, TS is the best method in SVHN with ResNet 152 (SD) model but its performance is very similar to the DIAG. The reason is that this model has a very high accuracy and the original model is actually already well calibrated. So, the single parameter TS would be enough to improve the calibration slightly. Our DIAG is the best method on datasets with larger number of classes and our OI is also comparable to it. Both these method have the best average ranking and DIAG has the best relative error on NLL.

Finally, Table 7 compares different methods in Classwise-ECE. While there is no single winning method on Classwise-ECE when the number of classes is less than 200, DIR is the best method on

Figure 6: Reliability diagrams and learned diagonal functions. See Fig. 4 for the explanation of each diagram and axis.

(a) Reliability Diagram  (b) Weighted Reliability Diagram  (c) Diagonal Functions

| Dir | MS | DIAG | OI | OP | Uncal. | TS |

Figure 7: Reliability diagrams and learned diagonal functions. See Fig. 4 for the explanation of each diagram and axis.

Figure 8: Accuracy, NLL, and ECE vs. calibration set size for CIFAR, CARS, BIRDS datasets. For each experiment, we use from $10\%$ to $100\%$ of the calibration set to train pos-hoc calibration functions and plot their accuracy, NLL, and ECE. Compared to DIR and MS, performance of the intra order-preserving methods (TS, DIAG, OI, and OP) degrades less with reducing the calibration set size.

Table 6: *NLL.*

| Dataset | Model | Uncal. | TS | DIR | MS | DIAG | OI | OP |
|---|---|---|---|---|---|---|---|---|
| CIFAR10 | ResNet 110 | $0.35827_7$ | $0.20926_5$ | $0.20511_3$ | $\mathbf{0.20375}_1$ | $0.20674_4$ | $0.20488_2$ | $0.20954_6$ |
| CIFAR10 | Wide ResNet 32 | $0.38170_7$ | $0.19148_3$ | $0.18203_2$ | $\mathbf{0.18165}_1$ | $0.19221_5$ | $0.19164_4$ | $0.19332_6$ |
| CIFAR10 | DenseNet 40 | $0.42821_7$ | $0.22509_3$ | $0.22371_2$ | $\mathbf{0.22240}_1$ | $0.22551_4$ | $0.23097_6$ | $0.22798_5$ |
| SVHN | ResNet 152 (SD) | $0.08542_7$ | $\mathbf{0.07861}_1$ | $0.08038_5$ | $0.08100_6$ | $0.07887_2$ | $0.07992_3$ | $0.08010_4$ |
| CIFAR100 | ResNet 110 | $1.69371_7$ | $1.09169_4$ | $1.09607_5$ | $\mathbf{1.07370}_1$ | $1.10091_6$ | $1.07966_2$ | $1.08375_3$ |
| CIFAR100 | Wide ResNet 32 | $1.80215_7$ | $0.94453_3$ | $0.95288_6$ | $\mathbf{0.93273}_1$ | $0.94928_4$ | $0.94312_2$ | $0.95001_5$ |
| CIFAR100 | DenseNet 40 | $2.01740_7$ | $1.05713_2$ | $1.05909_3$ | $\mathbf{1.05084}_1$ | $1.05972_4$ | $1.06127_5$ | $1.07626_6$ |
| CARS | ResNet 50 (pretrained) | $0.32993_7$ | $0.31813_4$ | $0.32381_6$ | $0.31904_5$ | $\mathbf{0.31234}_1$ | $0.31593_2$ | $0.31793_3$ |
| CARS | ResNet 101 (pretrained) | $0.30536_7$ | $0.29329_3$ | $0.29714_4$ | $0.29788_5$ | $\mathbf{0.28573}_1$ | $0.28897_2$ | $0.30444_6$ |
| CARS | ResNet 101 | $0.61185_7$ | $0.58619_4$ | $0.59504_6$ | $0.58683_5$ | $\mathbf{0.57385}_1$ | $0.57774_2$ | $0.58319_3$ |
| BIRDS | ResNet 50 (NTSNet) | $0.74676_7$ | $0.56569_4$ | $0.61239_5$ | $0.63055_6$ | $0.54915_2$ | $\mathbf{0.54508}_1$ | $0.56288_3$ |
| ImageNet | ResNet 152 | $0.98848_7$ | $0.94208_4$ | $0.95081_5$ | $0.95786_6$ | $\mathbf{0.92553}_1$ | $0.92850_2$ | $0.93935_3$ |
| ImageNet | DenseNet 161 | $0.94395_7$ | $0.90928_5$ | $0.91214_6$ | $0.90578_3$ | $\mathbf{0.88937}_1$ | $0.89552_2$ | $0.90632_4$ |
| ImageNet | PNASNet5 large | $0.80240_7$ | $0.75761_6$ | $0.73955_5$ | $0.71522_4$ | $\mathbf{0.65550}_1$ | $0.65674_2$ | $0.69595_3$ |
| Average Relative Error | | $1.000_7$ | $0.766_4$ | $0.772_6$ | $0.768_5$ | $\mathbf{0.749}_1$ | $0.751_2$ | $0.765_3$ |

Table 7: *Classwise ECE.*

| Dataset | Model | Uncal. | TS | DIR | MS | DIAG | OI | OP |
|---|---|---|---|---|---|---|---|---|
| CIFAR10 | ResNet 110 | $0.09846_7$ | $0.04344_5$ | $0.03950_4$ | $0.03615_2$ | $0.03791_3$ | $\mathbf{0.03454}_1$ | $0.04435_6$ |
| CIFAR10 | Wide ResNet 32 | $0.09530_7$ | $0.04775_4$ | $0.02947_2$ | $\mathbf{0.02921}_1$ | $0.05462_6$ | $0.04747_3$ | $0.04918_5$ |
| CIFAR10 | DenseNet 40 | $0.11430_7$ | $0.03977_4$ | $0.03687_2$ | $\mathbf{0.03678}_1$ | $0.03877_3$ | $0.04575_5$ | $0.05182_6$ |
| SVHN | ResNet 152 (SD) | $0.01940_4$ | $0.01849_2$ | $0.01988_5$ | $0.02088_6$ | $\mathbf{0.01478}_1$ | $0.01858_3$ | $0.02128_7$ |
| CIFAR100 | ResNet 110 | $0.41644_7$ | $0.20095_3$ | $\mathbf{0.18639}_1$ | $0.20270_5$ | $0.21966_6$ | $0.19977_2$ | $0.20237_4$ |
| CIFAR100 | Wide ResNet 32 | $0.42027_7$ | $0.18573_4$ | $\mathbf{0.17951}_1$ | $0.17966_2$ | $0.18636_5$ | $0.19397_6$ | $0.18484_3$ |
| CIFAR100 | DenseNet 40 | $0.47026_7$ | $0.18664_3$ | $0.18630_2$ | $0.19112_5$ | $\mathbf{0.18614}_1$ | $0.19866_6$ | $0.18752_4$ |
| CARS | ResNet 50 (pretrained) | $0.17353_3$ | $0.18515_7$ | $0.17094_2$ | $0.18312_6$ | $\mathbf{0.16891}_1$ | $0.18217_5$ | $0.17567_4$ |
| CARS | ResNet 101 (pretrained) | $0.16503_4$ | $0.17186_6$ | $0.15914_2$ | $0.17405_7$ | $0.16692_5$ | $0.16434_3$ | $\mathbf{0.15509}_1$ |
| CARS | ResNet 101 | $0.26300_2$ | $0.27234_6$ | $0.26333_3$ | $0.27447_7$ | $0.26594_5$ | $0.26488_4$ | $\mathbf{0.25097}_1$ |
| BIRDS | ResNet 50 (NTSNet) | $0.24901_3$ | $0.26369_7$ | $\mathbf{0.22920}_1$ | $0.25639_6$ | $0.25073_5$ | $0.25069_4$ | $0.24031_2$ |
| ImageNet | ResNet 152 | $0.31846_7$ | $0.30886_4$ | $\mathbf{0.30061}_1$ | $0.30895_5$ | $0.31372_6$ | $0.30642_3$ | $0.30081_2$ |
| ImageNet | DenseNet 161 | $0.30992_7$ | $0.30309_5$ | $\mathbf{0.29403}_1$ | $0.29807_2$ | $0.30659_6$ | $0.30248_4$ | $0.29959_3$ |
| ImageNet | PNASNet5 large | $0.31356_7$ | $0.25587_6$ | $\mathbf{0.23797}_1$ | $0.24283_2$ | $0.25004_5$ | $0.24634_4$ | $0.24493_3$ |
| Average Relative Error | | $1.000_7$ | $0.752_6$ | $\mathbf{0.704}_1$ | $0.734_3$ | $0.729_2$ | $0.740_4$ | $0.743_5$ |

this metric in ImageNet and in overall. In the next section, we discuss a hidden bias in Classwise-ECE metric that might become problematic. It seems Classwise-ECE might promote uncertainty in the output regardless of the actual accuracy of the model. This suggests there might be more investigation required for this metric and a practitioner should be cautious about these numbers.

### E.1 Is Classwise-ECE a Proper Scoring Rule Calibration Metric?

It is known that ECE is not a proper scoring rule and thus there exist trivial solutions which yield optimal scores [9]. In this section, we show the same holds for Classwise-ECE metric. Classwise-ECE is "defined as the average gap across all classwise-reliability diagrams, weighted by the number of instances in each bin:

$$\text{Classwise-ECE} = \frac{1}{k}\sum_{j=1}^{k}\sum_{i=1}^{m}\frac{B_{i,j}}{n}|y_j(B_{i,j}) - \hat{p}_j(B_{i,j})| \qquad (4)$$

where $k$, $m$, $n$ are the numbers of classes, bins and instances, respectively, $|B_{i,j}|$ denotes the size of the bin, and $\hat{p}_j(B_{i,j})$ and $y_j(B_{i,j})$ denote the average prediction of class $j$ probability and the actual proportion of class $j$ in the bin $B_{i,j}$." [5].

While the above definition of Classwise-ECE intuitively makes sense, we show that this metric fails to represent the quality of a predictor in a common degenerate case e.g. in a balanced dataset with $k$ classes one could achieve a perfect Classwise-ECE by scaling down the logits with a large enough positive scalar. A large enough temperature value increases the uncertainty of the model and brings all the class probabilities close to $1/k$ while maintaining the accuracy of the model. As the result, in all the classwise-reliability diagrams every data point falls into the bin that contains confidence values around $1/k$. Since the dataset is balanced, the actual proportion of class $j$ in that bin will also be $1/k$ so the model exhibits a perfect Classwise-ECE.

Table 8: *Temperature scaling effect on Classwise-ECE. A large temperature value improves the Classwise-ECE in most of the cases. The subscript numbers represent the rank compared to the values in Table 7. We remark that the purpose of this experiment is not to improve the performance but rather highlight the need for studying Classwise-ECE metric in the future works.*

| Dataset | Model | Uncal. | Uncal./1000 |
|---|---|---|---|
| CIFAR10 | ResNet 110 | 0.09846 | $0.00021_1$ |
| CIFAR10 | Wide ResNet 32 | 0.09530 | $0.00126_1$ |
| CIFAR10 | DenseNet 40 | 0.11430 | $0.00143_1$ |
| SVHN | ResNet 152 (SD) | 0.01940 | $0.33123_8$ |
| CIFAR100 | ResNet 110 | 0.41644 | $0.00080_1$ |
| CIFAR100 | Wide ResNet 32 | 0.42027 | $0.00199_1$ |
| CIFAR100 | DenseNet 40 | 0.47026 | $0.00282_1$ |
| CARS | ResNet 50 (pretrained) | 0.17353 | $0.16048_1$ |
| CARS | ResNet 101 (pretrained) | 0.16503 | $0.16108_3$ |
| CARS | ResNet 101 | 0.26300 | $0.15067_1$ |
| BIRDS | ResNet 50 (NTSNet) | 0.24901 | $0.05831_1$ |
| ImageNet | ResNet 152 | 0.31846 | $0.11074_1$ |
| ImageNet | DenseNet 161 | 0.30992 | $0.11074_1$ |
| ImageNet | PNASNet5 large | 0.31356 | $0.10960_1$ |

We remark that this problem does not happen with ECE, because ECE is computed with regard to the *accuracy* of the bins. While all the data points still fall inside the bin that contains the confidence value $1/k$, the accuracy of this bin would be equal to the accuracy of the model. Thus, there would be mismatch between the confidence and the accuracy of the bin, which results to a high ECE.

To validate this insight, we scale down the uncalibrated logit values by a large scalar number and see how it affects Classwise-ECE in Table 8. It shows this simple hack drastically improves the Classwise-ECE value of the uncalibrated models and outperforms the methods in Table 7 by large margin in most of the cases. Note that we can not achieve perfect Classwise-ECE because the datasets are not perfectly balanced.

We are concerned that this issue with *Classwise-ECE might bias future work to lean towards merely increasing the uncertainty of predictions without actually calibrating the model in a meaningful way.* To avoid this, Classwise-ECE metric should be always used with other proper scoring rule metrics (e.g., NLL or Brier) in evaluation. As we discuss in the next section, this issue would not happen when bins are dynamically chosen to ensure the number of data points in each bin remains equal.

## E.2 Debiased ECE and a Fix to Classwise-ECE

We believe that the issue mentioned above is due to the binning scheme used in estimating Classwise-ECE which allows all the data points fall into a single bin. Nixon *et al.* [8] propose an adaptive binning scheme that guarantees the number of data points in each bin remains balanced; therefore, it does not exhibit the same issue as Classwise-ECE. In addition to the binning scheme, Kumar et al. [6] introduce *debiased ECE* and multiclass *marginal calibration error* metrics that are debiased versions similar to the ECE and Classwise-ECE metrics, respectively. The idea is to subtract an approximate correction term to reduce the biased estimate of the metrics. For the completeness, we present *debiased ECE* and multiclass *marginal calibration error* for all the methods in Table 9 and Table 10, respectively. While the results in debaised ECE are similar to ECE, comparing the results in Table 7 and Table 10 shows DIAG is performing better in terms of multiclass marginal calibration error and outperforms DIR in average relative error.

Overall, although the intra order-preserving models are the winning methods among most of the ever-increasing calibration metrics, one should carefully pick the calibration method and the metric depending on their application.

Table 9: *Debiased ECE [6].*

| Dataset | Model | Uncal. | TS | DIR | MS | DIAG | OI | OP |
|---|---|---|---|---|---|---|---|---|
| CIFAR-10 | ResNet 110 | $0.09070_7$ | $0.01924_4$ | $0.01927_5$ | $0.01716_3$ | $0.01573_2$ | $\mathbf{0.00000_1}$ | $0.02282_6$ |
| CIFAR-10 | Wide ResNet 32 | $0.08661_7$ | $0.00809_2$ | $0.00943_3$ | $0.00958_4$ | $0.01717_6$ | $\mathbf{0.00194_1}$ | $0.01073_5$ |
| CIFAR-10 | DenseNet 40 | $0.10340_7$ | $0.01195_2$ | $0.01228_3$ | $0.01266_4$ | $\mathbf{0.01130_1}$ | $0.02410_6$ | $0.02309_5$ |
| SVHN | ResNet 152 (SD) | $0.01922_5$ | $0.00892_2$ | $0.00979_4$ | $0.00939_3$ | $\mathbf{0.00617_1}$ | $0.02269_6$ | $0.03429_7$ |
| CIFAR-100 | ResNet 110 | $0.22699_7$ | $0.02004_2$ | $0.02842_3$ | $0.03054_5$ | $0.05596_6$ | $\mathbf{0.00626_1}$ | $0.02903_4$ |
| CIFAR-100 | Wide ResNet 32 | $0.24827_7$ | $0.01031_2$ | $0.01909_5$ | $0.03018_6$ | $0.01498_4$ | $\mathbf{0.00545_1}$ | $0.01408_3$ |
| CIFAR-100 | DenseNet 40 | $0.26523_7$ | $\mathbf{0.00000_1}$ | $\mathbf{0.00000_1}$ | $0.02809_6$ | $\mathbf{0.00000_1}$ | $0.00432_4$ | $0.01265_5$ |
| CARS | ResNet 50 (pre) | $0.02327_6$ | $0.00900_2$ | $0.02512_7$ | $0.01605_4$ | $\mathbf{0.00000_1}$ | $0.01611_5$ | $0.01363_3$ |
| CARS | ResNet 101 (pre) | $0.02181_6$ | $0.01956_3$ | $0.02419_7$ | $0.01504_2$ | $0.01964_4$ | $0.02136_5$ | $\mathbf{0.01271_1}$ |
| CARS | ResNet 101 | $0.04280_5$ | $0.02654_3$ | $\mathbf{0.01518_1}$ | $0.03728_4$ | $0.02542_2$ | $0.04766_6$ | $0.04821_7$ |
| BIRDS | ResNet 50 (NTS) | $0.47117_7$ | $0.04054_4$ | $0.05545_5$ | $0.07224_6$ | $\mathbf{0.01518_1}$ | $0.01650_2$ | $0.03104_3$ |
| ImageNet | ResNet 152 | $0.07745_7$ | $0.02157_4$ | $0.05247_5$ | $0.06099_6$ | $\mathbf{0.00066_1}$ | $0.00941_2$ | $0.01804_3$ |
| ImageNet | DenseNet 161 | $0.06598_7$ | $0.02008_4$ | $0.04542_5$ | $0.04888_6$ | $\mathbf{0.00998_1}$ | $0.01158_2$ | $0.01924_3$ |
| ImageNet | PNASNet5 large | $0.06820_6$ | $0.09620_7$ | $0.05728_5$ | $0.03580_4$ | $0.01273_3$ | $\mathbf{0.00713_1}$ | $0.01272_2$ |
| Avgerage Relative Error | | $1.000_7$ | $0.357_3$ | $0.430_6$ | $0.409_5$ | $\mathbf{0.213_1}$ | $0.337_2$ | $0.406_4$ |

Table 10: *Marginal Calibration Error [6].*

| Dataset | Model | Uncal. | TS | DIR | MS | DIAG | OI | OP |
|---|---|---|---|---|---|---|---|---|
| CIFAR-10 | ResNet 110 | $0.00859_7$ | $0.00305_2$ | $0.00371_6$ | $0.00363_5$ | $0.00346_4$ | $\mathbf{0.00218_1}$ | $0.00336_3$ |
| CIFAR-10 | Wide ResNet 32 | $0.01516_7$ | $0.01408_3$ | $\mathbf{0.00410_1}$ | $0.00432_2$ | $0.01442_6$ | $0.01416_5$ | $0.01410_4$ |
| CIFAR-10 | DenseNet 40 | $0.01132_7$ | $0.00602_4$ | $\mathbf{0.00417_1}$ | $0.00583_2$ | $0.00601_3$ | $0.00729_6$ | $0.00686_5$ |
| SVHN | ResNet 152 (SD) | $0.00227_2$ | $0.00245_3$ | $0.00426_5$ | $0.00541_6$ | $\mathbf{0.00178_1}$ | $0.00387_4$ | $0.00691_7$ |
| CIFAR-100 | ResNet 110 | $0.00315_7$ | $\mathbf{0.00129_1}$ | $0.00185_5$ | $0.00233_6$ | $0.00144_3$ | $0.00141_2$ | $0.00151_4$ |
| CIFAR-100 | Wide ResNet 32 | $0.00356_7$ | $0.00266_4$ | $0.00222_2$ | $\mathbf{0.00199_1}$ | $0.00270_6$ | $0.00268_5$ | $0.00257_3$ |
| CIFAR-100 | DenseNet 40 | $0.00417_7$ | $0.00266_6$ | $\mathbf{0.00222_1}$ | $0.00263_5$ | $0.00261_4$ | $0.00259_3$ | $0.00234_2$ |
| CARS | ResNet 50 (pre) | $0.00063_6$ | $0.00058_2$ | $\mathbf{0.00035_1}$ | $0.00090_7$ | $0.00060_5$ | $0.00059_3$ | $0.00059_3$ |
| CARS | ResNet 101 (pre) | $0.00043_3$ | $0.00044_4$ | $0.00044_4$ | $0.00092_7$ | $0.00041_2$ | $\mathbf{0.00034_1}$ | $0.00046_6$ |
| CARS | ResNet 101 | $\mathbf{0.00114_1}$ | $0.00114_1$ | $0.00173_6$ | $0.00230_7$ | $0.00114_1$ | $0.00118_5$ | $0.00117_4$ |
| BIRDS | ResNet 50 (NTS) | $0.00934_7$ | $0.00139_4$ | $0.00148_6$ | $0.00141_5$ | $0.00138_3$ | $0.00132_2$ | $\mathbf{0.00130_1}$ |
| ImageNet | ResNet 152 | $0.00040_6$ | $0.00038_2$ | $\mathbf{0.00034_1}$ | $0.00042_7$ | $0.00038_2$ | $0.00038_2$ | $0.00038_2$ |
| ImageNet | DenseNet 161 | $0.00041_7$ | $0.00039_3$ | $\mathbf{0.00035_1}$ | $0.00038_2$ | $0.00039_3$ | $0.00039_3$ | $0.00039_3$ |
| ImageNet | PNASNet5 large | $0.00039_7$ | $0.00032_6$ | $\mathbf{0.00025_1}$ | $0.00028_2$ | $0.00028_2$ | $0.00029_4$ | $0.00030_5$ |
| Average Relative Error | | $1.000_7$ | $0.750_3$ | $0.735_2$ | $0.996_6$ | $\mathbf{0.725_1}$ | $0.778_4$ | $0.898_5$ |

## Footnotes

[4]This choice is only for convenience of writing the indices.

[5]https://github.com/markus93/NN_Calibration

[6]https://github.com/osmr/imgclsmob/blob/master/pytorch/README.md