[Reviews · NeurIPS 2020]

Review 1

Summary and Contributions: This paper proposes a general post-hoc calibration function class for multi-class deep classifiers. The motivation for the paper is that the current calibration approaches lack sufficient representation or lead to degenerate accuracy. To remedy the issue, authors propose a learnable space of functions called intra order-preserving functions and their two variants to tackle the problem. They empirically evaluated their results on multiple standard datasets.

Strengths: This paper Introduces intra order-preserving calibration functions that preserve top-k classification accuracy. The order-preservation or ranking-based calibration intuitively makes sense and is suitable for the problem. I appreciate the rigour that the authors aim at providing to the problem. S2. Experiments outperform state-of-the-art related methods.

Weaknesses: While I appreciate that the authors seek a general class of functions to tackle the problem (and I fully support that), the specific of what is different from what is proposed and a conventional ranking-based approach is not clear. Paper introduces notations and theorems up to section 3.4, and then suddenly a simple approach is proposed that does not seem to be clearly linked to the rest of the paper. If paper started with the simplified architecture and supporting experiments, then it would have a clearer paper to read and understand. The classwise-ECE metric was originally given in [14] and DIR improved it The proposed methods can't outperform DIR in this metric, and the reason is not clear. The difference between these methods and scope of usage should be clearly discussed. Table 1 in the experiment section doesn't show accuracy for the proposed methods, but that can be helpful when compared with previous works in terms of network calibration. A better calibration method shouldn't influence accuracy as described in this paper. More experiments are required to demonstrate how the intra order-preserving functions preserve top-k prediction accuracy. Since authors claim a family, which to me requires experiments with various members of the family. Authors, however, seem to have fixed a single choice of architecture and experimented with the sub-family members (which is great but not complete for the claim made). It is not clear also how to create variations of the sub-families with the fixed architecture. The final message of the paper is not clear to me: which architecture/family should I choose if I am to use the proposed approach?

Correctness: To the best of my understanding, it seems correct

Clarity: The paper could be improved in my opinion, in the sense that the flow and what each part of the proposed approach seeks to achieve is unclear. There are some notations that are introduced and later seemed to be changed (w -> m) or some that are not clear why introduced at all (e.g. sigma). Figure 2 that was supposed to clarify the flow, introduces confusion: for instance, the "order invariant?" box

Relation to Prior Work: Partially. The proposed methods can't outperform DIR in Classwise-ECE metric, the difference between these methods and scope of usage should be clearly discussed.

Reproducibility: Yes

Additional Feedback: With much discussion about the paper, I still feel that the paper could be better presentated and some of the notations streamlined for better adoption in the community. At the same time, I appreciate the technique employed by the authors and the contributions made (even though larger scale experiments could really help). Considering the author's feedback and discussions with other reviewers, I am happy to change my score with the hope that the authors reflect on some of the comments in their camera-ready submission and improve the presentation further.


Review 2

Summary and Contributions: The paper introduces a learnable space of functions, called intra order-preserving functions to preserve top-k predictions in post-hoc calibration. Additionally, order-invariant and diagonal structures are also introduced to improve generalization. The paper studies the properties and proposes a neural network architecture to represent these functions. Experiments on benchmark datasets are conducted to demonstrate the effectiveness of the proposed method regarding various calibration performance metrics.

Strengths: The paper provides both solid theoretical analysis and empirical implementation. The proposed method is well-motivated with a clear introduction of background and existing problems. Using the proposed new family of functions generalizes many existing calibration techniques. As far as I know, the paper provides novel contributions to neural network calibration.

Weaknesses: It would be better if visualizations of the learned calibration functions could be presented in the experiments, which helps interpretate where the improvements come from.

Correctness: I checked the definations and theorems in the main paper and did not find mistakes but I did not look into the proofs carefully in the supplementary material.

Clarity: The paper is well written and clear. Minor comments: Line 287 the number of classes vary from → varies

Relation to Prior Work: The relation to prior work on many post-hoc calibration methods is discussed in detail. More previous work on learning order-preserving functions with neural networks should be included in the paper.

Reproducibility: Yes

Additional Feedback: Post author feedback: Thanks the authors for the detailed rebuttal. After reading other reviews and discussions, I think this is a strong paper and will maintain my original score.


Review 3

Summary and Contributions: A new classifier calibration method is proposed for deep neural nets. The key idea is to come up with a method which ensures that the class predictions remain exactly the same, by using intra order-preserving functions. These functions are implemented to be differentiable and thus usable within a neural net. The results show improvements over the existing methods.

Strengths: * The paper is well written. * It contains both theoretical and practical contributions, all very valuable for the community. * The experimental results show significant improvements over the state-of-the-art in ECE, some improvements in NLL and marginal calibration error. * Relevant related work has been covered very well. * Source code has been provided.

Weaknesses: * In the main paper it has not been discussed why the quality of the proposed methods is quite different for ECE, NLL, classwise-ECE. I acknowledge the space constraints, but a few sentences should be added. * The arguments in the discussion about evaluation measures in the supplemental material are not convincing (see details below).

Correctness: Yes, all seems correct.

Clarity: The paper is well written.

Relation to Prior Work: Yes, it is clear how this work differs from previous contributions.

Reproducibility: Yes

Additional Feedback: The proposed methods perform very strongly in ECE, slightly better than the state-of-the-art in NLL and slightly worse in classwise-ECE. It would be good to have some explanation about why ECE and classwise-ECE give so different results. As ECE studies the calibration of only the class with the highest predicted probability and ignores other class probabilities, does it mean that the proposed method is better than the state-of-the-art in top-1 probability but slightly weaker on other classes? In the appendix provided as supplemental material, at lines 739-742 it is claimed that ECE does not suffer from the same problem that is highlighted about classwise-ECE at lines 731-738. While this is technically correct, it misses the point. Actually, ECE also suffers from essentially the same problem. It is very easy to modify predictions such that the accuracy would remain the same and ECE would get down close to zero. However, instead of using temperature scaling with a high temperature, one should change in each prediction the highest probability into the real number representing the value of accuracy and then distribute the change among all other classes evenly, to ensure that the probability vector still adds up to 1. After doing so, all predictions are in the same bin (the bin which contains the real number representing the classifier's accuracy) and the average confidence is in perfect agreement with the actual accuracy. Indeed, one needs to know the value of accuracy for this, and in this sense ECE is slightly harder to fool than classwise-ECE, but it still implies that ECE alone is not a sufficient metric. Therefore, ECE must always be evaluated in combination with some other measure, such as NLL, which would detect such kind of fooling. Similarly, any attempt to fool classwise-ECE would also be detected by deteriorating NLL. Since NLL is roughly comparable for all methods except 'Uncal', then the discussion in the appendix Section D.1 does not provide an explanation to why the proposed methods are not the best at classwise-ECE. I suggest the authors to either drop D.1 and Table 9, or provide a similar 'fooling' example for ECE also. Additionally, it should emphasized that it is important to complement ECE or classwise-ECE with an additional measure, preferrably a proper scoring rule, such as NLL or Brier score. Note that the word 'proper' in the title of D.1 at line 725 of the supplementary is confusing, because it could be interpreted as the 'proper scoring rule', see https://en.wikipedia.org/wiki/Scoring_rule#Proper_scoring_rules --- After rebuttal: Thank you very much for doing a very good job addressing our concerns in the rebuttal!


Review 4

Summary and Contributions: This paper deals with post-hoc calibration of predictions from deep neural networks. The authors argue that for two-step calibration methods, post-hoc learning of a mapping of the network output into a calibrated confidence score, the validation set used for calibration is not sufficient for modeling properly the landscape of these predictions and ends up overfitting and reducing the original accuracy of the network. In this context they study and propose approaches that do not alter the order of the predictions, i.e. no loss in accuracy, which are here called _intra order-preserving_. The authors argue and model such approaches along with a few extra regularization structures that leverage patterns from different input components (classes). The proposed structures are here termed _order-invariant_ (swapping elements of the input vectors leads to swapping of the same elements in the output calibrated vector) and _diagonal_ (different class predictions do not interact with each other in the calibration function). The author study multiple variants of calibration methods that mix 2 or more of the mentioned structures. In practice this leads to mostly straight-forward implementations as post-hoc calibration networks. The authors evaluate the proposed solutions across several datasets and architectures with encouraging results. ---------------------- [Post-rebuttal update] I thank the authors for the rebuttal that offered a nice toy example and additional results showing the interest of using these techniques with ever smaller calibration sets. After checking out this rebuttal and feedback from other reviews, I think this is a strong paper. For the final version of the paper it would be nice if the authors showed results on OOD detection or dataset shift, without using an OOD dataset during training. For instance they can show the evolution of the calibration scores of the model by directly testing on CIFAR-C or ImageNet-C (Hendrycks and Dietterich (ICLR 2019)) as evaluated by Ovadia et al. (NeurIPS 2019).

Strengths: - The paper addresses a highly important topic for using deep neural networks in practice - The ideas in the paper bring novelty. The formalism proposed by the authors to connect the various sub-families of function spaces for mapping the original logits is quite good and clear. It is generic and can incorporate other methods in the area. I consider it as a contribution of this paper. In general the arguments brought by the authors are sound and further detailed in the supplementary when needed. The paper is mostly well organized, with a few exceptions. - The authors benchmark their contributions across several datasets and architectures (this also due to an advantage of the method itself as it is trained directly on logits and training times are short) - The discussion in the appendix regarding the flaws of the class-wise ECE compared to ECE is also useful and good. It might have been better placed in the main paper but the current format of the paper doesn't allow it. The authors provide other useful discussions regarding the metrics used for this task. - The code provided in the supplementary is clear and easy to use and the authors follow sane evaluation protocols, k-fold cross-validation over the validation sets of the considered datasets.

Weaknesses: - Although the proposed formalism is nice, it can be sometimes dry and confusing, especially for less experience people in the area, giving the impression of over-complicating things, especially since the implementation is more straight forward. I would suggest further working on section 3 to clarify the context of each sub-family, further connections with other methods (to the merit of the authors they mention the connections with temperature scaling and softmax) and coming back to the big picture at intermediate points (e.g. before introducing order-invariant). A toy example with some visualizations on prediction evolutions on the simplex in the style from Vaicenavicius et al. [i], Kull et al. [14] might help. - The authors argue that the calibration dataset is usually small and that this leads to the calibration module to overfit easily and reducing the accuracy of the network. However, unless I'm mistaken, there are no empirical results showing this. It would be useful to see how are the proposed contributions coping with different sizes of the calibration set. In a slightly related work on failure prediction [ii], Corbiere et al. argued that an issue with smaller calibration set is in fact related to the smaller amount of wrongly predicted samples that would be needed to properly tune the calibration across prediction intervals. They show eventually that using the train set helps a bit more. Back to this paper, it might be useful to show the resilience of these methods to varying calibration set sizes. - I appreciate that the authors go beyond the usual CIFAR-10/100 datasets for evaluating calibration, however they all concern regular image classification, albeit with different complexities in class number. Another key task for calibration is out-of-distribution detection (which can a more difficult test) and I consider it would further emphasize the advantages of this work. There are a few popular benchmarks: CIFAR-10C [iii], using SVHN as OOD or other datasets [29]. Minor - The authors point out the limitations of class-wise ECE, and in the end use rather ECE, Brier and NLL. Recent papers argue for other meaningful calibration metrics that are adaptive (ACE) or thresholded (TACE) (Nixon et al.[24]) and could be analysed here. - The implementation of the diagonal intra order-preserving version is very briefly described in the paper, while it requires an Unconstrained Monotonic Neural Network [30] and that is not mentioned further either. I would suggest adding more details and context on what UMNN's do here, their complexity, etc. References: [i] J. Vaicenavicius et al., Evaluating model calibration in classification, PMLR 2019 [ii] C. Corbiere et al., Addressing Failure Prediction by Learning Model Confidence, NeurIPS 2019 [iii] Y. Ovadai et al., Can you trust your model’s uncertainty? Evaluating predictive uncertainty under dataset shift, NeurIPS 2019

Correctness: The paper is technically sound. Experiments are correctly conducted.

Clarity: The paper is mostly clearly written.

Relation to Prior Work: Yes, the author clearly discuss relation to previous works.

Reproducibility: Yes

Additional Feedback: Minor comments: - ResNet-152 is quite a big beast for a thumbnail image dataset as SVHN. - x = g(z) is introduced only in line 238 while x is used multiple times before that. It would be better introduced earlier. - the names of the "sm" and "m" functions might be confusiong. Maybe some other distinct notations would avoid confusion. Suggestions: - Here a are few suggestions for improving the paper and that could be eventually addressed in the rebuttal, if time allows it: + could the authors mention how are the methods performing on varying sized of the calibration dataset + could the authors discuss how are methods doing on OOD tasks. Conclusion: - This work is interesting and convincing. There are some parts that could be further improved or polished, but my current recommendation for this work is towards acceptance.

[Author Response · NeurIPS 2020]

We thank the reviewers for the insightful comments. Due to space limitation, we only discuss major comments below.

**R1,2,5 Intuitive Example.** We will add a toy example suggested by R5 at the beginning of Sec. 3 to visualize different

post-hoc calibration functions. This example is shown in Fig(a) below. Here we visualize OP and OI models trained

using the same experiment setting as Fig. 2 of Kull *et al.* [14] on the 3-class Abalone UCI dataset. Each colored subset in

the simplex denotes a region with the same input order (i.e., class prediction order); e.g., inside the red region, we have

$x_3 > x_2 > x_1$. Each arrow depicts how an input is mapped by a trained calibration function. Unlike UNCONSTRAINED

model that can freely map the input probabilities and possibly change the order and accuracy, OP enforces the outputs

to stay within the same colored region as the inputs, but the vector fields can be different across regions. OI further

keeps the function permutation invariant, enforcing the vector fields to be the same among all the 6 colored regions (as

reflected in the symmetry in the visualization). This invariance property of OI can significantly reduce the hypothesis

space in learning, from the functions on whole simplex to functions on one colored region, for better generalization.

DIAG figure is visually similar to OI since there are only 3 classes and is not shown due to the space limitation.

**R1,3, 5 On Classwise-ECE Metric.** We first note that the optimal score for classwise-ECE does not necessarily correspond

to a perfect prediction as there are trivial solutions which yield optimal scores; therefore, as mentioned by R3, classwise-

ECE must always be evaluated along with other proper scoring rules, such as NLL and Brier, for a meaningful

comparison. This has been shown for ECE (e.g., Sec. 3 of [i], pointed out by R3), and we show the same applies to

classwise-ECE in the Appendix Sec. D.1. Given this issue, DIR does not perform well overall: while DIR is superior

in classwise-ECE, it is not good in NLL (Ranked 6th in Table 2) and Brier (Ranked 3rd in Table 6 in the Appendix),

both of which are proper scoring rules [i]. To further understand this, in Sec. D.2 we evaluate the performance of all

methods in terms of classwise Marginal Calibration Error (MCE) metric of [ii]. This metric is a debiased version of

classwise-ECE metric and does not suffer from the fooling example in Sec. D.1 due to its adaptive binning scheme (see

detailed discussion in D.1 and D.2). As Table 11 shows, DIAG has the best overall performance in this metric. Looking

at NLL, Berier, and MCE metrics the hypothesis (brought up by R3) that our method is better on top-1 prediction while

being weaker for other classes is not supported. Nonetheless, we acknowledge that future research is required to better

understand the performance difference between classwise-ECE and other classwise metrics like Brier and MCE.

**R1.** • Showing accuracy or top-k in Table 1 is redundant since they are the same as the uncalibrated network; the

proposed method is designed to enforce this constraint (which we verified in all experiments). • We are not limited to a

single architecture, as cross-validation was employed for architecture search (see L298-300); Table 5 in the Appendix

shows the selected architecture for each method. • In L215-216, **m** and $\sigma$ are *one* factorization of **w**, which we

introduced to ease the implementation within deep neural networks; we still use **w** in Theorem 1 to keep the framework

generic. • The order-invariant box in Fig. 2 denotes a decision box: if `No` (the order-preserving case), the network is fed

with the original input; if `Yes` (the order-invariant case), the sorted input is used; We will update this figure and use

the toy example above to build up the intuition before going to the math as the reviewer suggested. • *Main Message:*

Our paper introduces the order-preserving family to address the accuracy drop issue in using multilayer networks for

post-hoc calibration (see the performance of UNCONSTRAINED in Table 1). Searching inside this family allows one to

use complex post-hoc calibration functions without losing accuracy. While DIAG works well in most experiments here,

the best subfamily and architecture depend on many factors (e.g., the backbone model, metric, and calibration set size)

and need to be determined by cross-validation.

**R3.** We will update Sec D.1 as follows: Before giving the fooling example, we highlight that ECE is not a *proper*

*scoring rule* based on the definition in [i] and refer to Sec. 3 of [i] for an example; at the end, we emphasize that these

metrics should be used with other proper scoring rule metrics (e.g., NLL or Brier) in evaluation.

**R5.** • The UNCONSTRAINED results in Table 1 shows that a naive multi-layer perceptron can overfit. As requested, we

analyse the methods' resilience to varying calibration set size in Fig(b). The plots show DIAG and OI methods are more

stable when using a fraction of the calibration set ($x$-axis) compared to DIR and MS, highlighting the importance of

the order-preserving family in low data regimes. We observe a similar trend in other datasets/models and will include

these results in the Appendix. • We remark that the debiased ECE and classwise marginal calibration error metrics

[ii] illustrated in Sec D.2 of Appendix (see Table 10 and 11 in Appendix for the evaluation) use ACE in addition to a

debiasing technique to improve ECE and classwise-ECE, respectively. We will add a discussion about the importance

of TACE and ACE. • We were not able to finish the OOD experiments on time and have to do it in future work.

**References** [i] Y. Ovadai et al., Can you trust your model's uncertainty? Evaluating predictive uncertainty under dataset shift,

NeurIPS 2019. [ii] A. Kumar et al., Verified uncertainty calibration, NeurIPS 2019.

Fig(a) Learned calibration functions visualisation on simplex.

Fig(b) Performance vs. calibration set size

[Meta-Review · NeurIPS 2020]

The paper addresses the problem of post-training calibration in deep nets which has seen a lot of interest in the community lately. The work brings theoretical and practical contributions that are valuable to both researchers and practitioners. The proposed technique may also be useful for other problems besides post-training calibration. I would encourage the authors to try to improve the clarity of the presentation.